# NeuralCohort: Cohort-aware Neural Representation Learning for Healthcare Analytics

**Changshuo Liu** [1]  **Lingze Zeng** [1]  **Kaiping Zheng** [1]  **Shaofeng Cai** [1]  **Beng Chin Ooi** [1,2]  **James Wei Luen Yip** [3]

## Abstract

Electronic health records (EHR) aggregate extensive data critical for advancing patient care and refining intervention strategies. EHR data is essential for epidemiological study, more commonly referred to as *cohort study*, where patients with shared characteristics or similar diseases are analyzed over time. Unfortunately, existing studies on cohort modeling are limited, struggling to derive fine-grained cohorts or effectively utilize cohort information, which hinders their ability to uncover intrinsic relationships between cohorts. To this end, we propose NeuralCohort, a cohort-aware neural representation learning method that precisely segments patients into finer-grained cohorts via an innovative cohort contextualization mechanism and captures both intra- and inter-cohort information using a Biscale Cohort Learning Module. Designed as a plug-in, NeuralCohort integrates seamlessly with existing backbone models, enhancing their cohort analysis capabilities by infusing deep cohort insights into the representation learning processes. The effectiveness and generalizability of NeuralCohort are validated across extensive real-world EHR datasets. Experimental results demonstrate that NeuralCohort consistently improves the performance of various backbone models, achieving up to an 8.1% increase in AUROC.

## 1. Introduction

Electronic health records (EHR) are systematized collections of patients' medical histories, stored electronically within healthcare systems, which typically consist of patient demographics and temporal medical features (Johnson et al.,

[1]National University of Singapore, Singapore [2]Zhejiang University, China [3]National University Hospital, Singapore. Correspondence to: Changshuo Liu <shuo@nus.edu.sg>.

*Proceedings of the 42$^{nd}$ International Conference on Machine Learning*, Vancouver, Canada. PMLR 267, 2025. Copyright 2025 by the author(s).

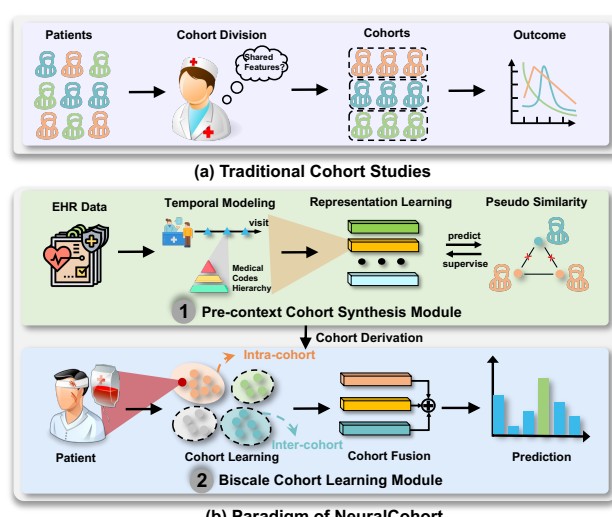

*Figure 1.* Traditional cohort studies and the paradigm of NeuralCohort to achieve interpretable cohort intelligence.

2016). EHR data is instrumental in advancing data-driven approaches within the healthcare sector and facilitating critical clinical decision-making (Yan et al., 2020; Wynants et al., 2020; Ooi et al., 2015) for more optimized patient management, including assisting doctors in assessing patients' health conditions, developing treatment plans, and proactively averting adverse outcomes with heightened efficacy and intelligence (Hou et al., 2020; Li et al., 2025; Lin et al., 2025), among others.

Contemporary approaches (Huang et al., 2019; Choi et al., 2017; Shang et al., 2019) primarily focus on extracting representations from the intrinsic information encapsulated within EHR data, which are subsequently used to enhance different downstream healthcare analytic tasks. Unfortunately, one indispensable consideration to achieve effective EHR representation learning has been largely overlooked by most existing studies - **Patient Cohorts**. In healthcare research, as depicted in Figure 1 (a), patient cohorts pertain to groups of individuals identified by shared characteristics, ranging from demographics to specific health conditions. Patient cohorts are integral to researching and developing effective medical interventions. For instance, when COVID-

19 ravages the world, doctors may opt to construct two distinct patient cohorts: one comprising COVID-19 patients aged between 20 and 40, and the other encompassing those aged between 50 and 70. Notably, elderly patients are more likely to exhibit severe symptoms such as breathing distress, cognitive decline, and elevated fever, while younger patients typically present with milder symptoms, such as fever, cough, and sore throat, or even remain asymptomatic (Alimohamadi et al., 2020). In addition, doctors may observe a consistent pattern among patients with fever symptoms, predominantly diagnosing them with COVID-19 (Viner et al., 2021). The former analysis indicates that the elderly population is prone to more severe symptoms, whereas the latter reveals a discernible association between fever and COVID-19, which serves as a valuable diagnostic indicator. Drawing from these insights, a significant optimization goal of EHR representation learning is to develop an effective cohort construction approach that advances from *individual intelligence* to *interpretable cohort intelligence*, thereby unleashing the profound potential of EHR data to facilitate EHR analysis and deepen the understanding of patient insights.

A straightforward approach to cohort division relies exclusively on explicit features and uses either a single feature or a combination of several features to derive cohorts across different healthcare analytic tasks. Despite being intuitive, this approach may not satisfactorily tackle EHR data in practice, as it fails to achieve the following two crucial desiderata: (1) **Fine-grained Cohort Division.** EHR data records patients' temporal visits to the hospital for different reasons, thereby engendering different diagnoses, medications, etc. Consequently, a coarse-grained cohort division may place similar patients into different cohorts at varying time points, potentially introducing noise into the representation learning process. This underscores the need for a fine-grained cohort division that takes into account the patient similarity in the time dimension, rather than relying solely on explicit features. (2) **Local Intra-cohort and Global Inter-cohort Information Exploitation.** As previously discussed, the similarity of patients within a cohort, *i.e.*, intra-cohort information, is valuable to medical decisions. Additionally, contrasting the characteristics of a patient against those from different cohorts, *i.e.*, inter-cohort information, can provide valuable insights and enhance diagnostic accuracy. Therefore, effective EHR representation learning should comprehensively and jointly leverage both intra- and inter-cohort information to utilize EHR data to the utmost. However, existing methods fail to simultaneously fulfill both desiderata.

To this end, we propose NeuralCohort, a cohort-aware neural representation learning method for healthcare analytics. As illustrated in Figure 1 (b), NeuralCohort is structured into two key modules. **Pre-context Cohort Synthesis Module**:

an innovative cohort synthesis task to model the temporal EHR data with pseudo patient similarity, thereby paving the way for constructing fine-grained cohorts. **Biscale Cohort Learning Module**: synergistically distill the cohort information both from local intra-cohort and global inter-cohort perspectives and finally derives the overall cohort information to enhance EHR data representation learning. Our contributions are summarized as follows.

- We propose NeuralCohort, a cohort-aware neural representation learning method to support fine-grained cohort generation and simultaneously exploit local intra-cohort and global inter-cohort information, which have been overlooked by prior studies in EHR analysis.

- We design a two-module paradigm for NeuralCohort. In Pre-context Cohort Synthesis Module, we model patients' temporal EHR data with pseudo patient similarity to construct fine-grained cohorts. Subsequently, in Biscale Cohort Learning Module, we capitalize on local intra-cohort and global inter-cohort information and encode them into augmented representations for prediction.

- NeuralCohort can be seamlessly integrated with various backbone models, serving as a versatile plug-in to incorporate cohort information into healthcare analytics, thereby enhancing overall performance.

- We evaluate the effectiveness of NeuralCohort on three real-world EHR datasets across two tasks. The experimental results demonstrate that NeuralCohort, when integrated with backbone models, outperforms its counterparts by a large margin of up to 8.1% in AUROC, confirming the efficacy of the comprehensive cohort information learned by NeuralCohort.

## 2. Related Work

EHR data encompasses a wide array of information, including structured data such as lab results and medications, as well as unstructured data such as clinical notes (Sauer et al., 2022). Employing representation learning methods is crucial for comprehending the intricate relationships inherent within heterogeneous EHR data (Landi et al., 2020; Cai et al., 2021; Steinberg et al., 2021; Zheng et al., 2021; 2022; Cai et al., 2022; Cheng et al., 2016; Su et al., 2025). For instance, CompNet (Wang et al., 2019) uses a graph convolutional model, enhanced by a reinforcement learning method, to learn the interactions between different medications and predict medication combinations. Beyond deriving representations from the EHR data directly, several studies integrate patient similarity to enhance the efficacy of representation learning methods. A cosine similarity-based patient graph is constructed to aggregate the patient similarity information

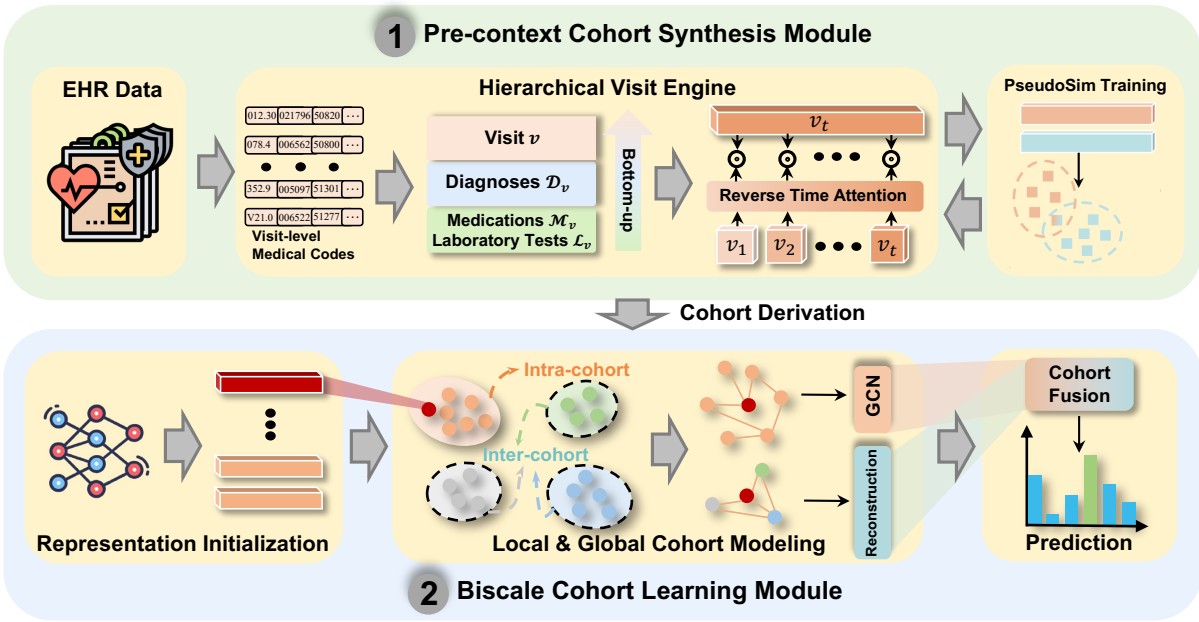

*Figure 2.* Overview of NeuralCohort. NeuralCohort first models patients' visit-level representations to derive patient-level representations under a PseudoSim Training task. The derived representations are used to generate cohorts. NeuralCohort fuses the representations of the intra- and inter-cohort graphs with those derived from the backbones for the final prediction.

for enhancing length of stay prediction (Zang et al., 2023). GRASP (Zhang et al., 2021) introduces a framework that characterizes patient similarity through learned inter-cohort representations for various downstream clinical tasks. Unfortunately, these methods predominantly focus on EHR data without exploiting the potential impact of the similarity relationships among patients on EHR representation learning. Further, some restrict their analytics to cohort-level similarity, and hence fall short of fully capturing the complex similarity relationships among patients. NeuralCohort mitigates the problem by generating fine-grained cohorts and leveraging cohort information from both intra- and inter-cohort perspectives. This dual-module method allows for a more nuanced understanding and utilization of cohort dynamics, significantly enhancing the effectiveness of EHR data representation learning.

There have been patient similarity prediction proposals (Sun et al., 2012; Suo et al., 2018; Navaz et al., 2022). For instance, a Mahalanobis distance learning method (Wang & Sun, 2015) is proposed to integrate human experts' knowledge and patients' historical data to measure patient similarity. NeuralCohort differs from these works, which typically leverage deep learning models to discern intricate patterns among patients for deriving similarity, in that we employ pseudo patient similarity to drive the generation of cohorts. This difference enhances NeuralCohort's ability to integrate and leverage cohort dynamics for more robust and contextu-

ally aware EHR representation learning.

For a more comprehensive discussion on traditional cohort studies and deep clustering, please refer to Appendix B.

## 3. Methodology

### 3.1. Problem Formulation

Given an EHR dataset comprising $N^p$ patients $\mathcal{P} = \{p_1, p_2, \cdots, p_{N^p}\}$, each patient $p$ has a temporal sequence of visits $\mathcal{V}_p = \{v_1, v_2, \cdots, v_{N^v}\}$, where $N^v$ is the number of visits for patient $p$. The features of each visit $v$ encompass three categories of medical codes: diagnoses $\mathcal{D}_v = \{d_1, d_2, \cdots, d_{N^d}\}$, medications $\mathcal{M}_v = \{m_1, m_2, \cdots, m_{N^m}\}$, and laboratory tests $\mathcal{L}_v = \{l_1, l_2, \cdots, l_{N^l}\}$, where $N^d$, $N^m$ and $N^l$ denote the numbers of diagnosis codes, medication codes and laboratory test codes associated with visit $v$, respectively.

Given a backbone model, our objective is to enhance the representation learning of EHR data by incorporating patient cohorts. To attain this goal, we propose NeuralCohort, a cohort-aware neural representation learning method for healthcare analytics. As depicted in Figure 2, we present the training pipeline $M(\mathcal{P}; \Theta_P, \Theta_\mathcal{C})$ as follows:

$$\underbrace{M(\mathcal{P}; \Theta_P, \Theta_C)}_{\text{NeuralCohort}} = \underbrace{M_P(\mathcal{S}(\mathcal{P})|\mathcal{P}; \Theta_P)}_{\text{Pre-context Cohort Synthesis}} \rightarrow \underbrace{M_C(\mathcal{C}, R_0; \Theta_C)}_{\text{Biscale Cohort Learning}} \quad (1)$$

where $M_P$ and $M_C$ correspond to the modules for the Pre-

context Cohort Synthesis and Biscale Cohort Learning, respectively, with associated parameters $\Theta_P$ and $\Theta_C$. The Pre-context Cohort Synthesis Module aims to derive cohorts $\mathcal{C}$ via modeling pair-wise pseudo patient similarity $\mathcal{S}(\mathcal{P})$. The Biscale Cohort Learning Module leverages these cohorts $\mathcal{C}$ alongside the representations from the backbone $\mathcal{R}_0$, resulting in enhanced representations.

### 3.2. Pre-context Cohort Synthesis Module

The Pre-context Cohort Synthesis Module is geared towards learning patient-level representations and consequently deriving cohorts. In essence, NeuralCohort first constructs a hierarchical network tailored for processing medical codes to extract visit-level features $\mathcal{R}_v$ from the EHR data. Subsequently, NeuralCohort proceeds to learn the patient-level features $\mathcal{R}_p$ based on both the visit-level features $\mathcal{R}_v$ and the pair-wise pseudo patient-level similarity $\mathcal{S}(\mathcal{P})$, serving as the basis for Biscale Cohort Learning.

**Hierarchical Visit Engine.** Each visit $v$ contains a set of diagnosis codes $\mathcal{D}_v$, medication codes $\mathcal{M}_v$ and laboratory test codes $\mathcal{L}_v$. Notably, diagnosis codes adhere to specific ontologies that standardize the classification and interpretation of medical data. For example, the ICD-9 (International Classification of Diseases, Ninth Revision) employs a tree-like structure where each leaf corresponds to a standardized diagnosis code and a unique medical term (Quan et al., 2005). The leaves and their respective ancestors delineate hierarchical paths within the ontology. A specific instance of path $Root \rightarrow 240\text{–}279 \rightarrow 249\text{–}259 \rightarrow 250 \rightarrow 250.4$ represents the hierarchical path of the leaf *250.4*, with *250* designating *Diabetes mellitus* and *250.4* denoting *Diabetes with renal manifestations*. To distinguish leaves with analogous structures and derive informative diagnosis code representations, we partition the ontology into two constituents:

- **Path.** This component traces the path in the ontology tree from the *Root* to a leaf. We construct a graph based on the tree structure of the ontology and employ a graph-based method to obtain the representation of the path. However, certain leaves, such as *250.4* and *250.5*, may share similar paths, which necessitates additional differentiation strategies.

- **Semantics.** Each leaf within the ontology embodies a unique medical term. Ancestors convey board semantics, whereas leaves provide specific semantic details. Further, the semantic information inherent in the same ancestor can vary for different children. To tackle this, we adopt a semantic similarity metric between the text representations of an ancestor and a leaf as the weight for aggregating their representations. This combined consideration of paths and semantics enables the differentiation of individual leaves.

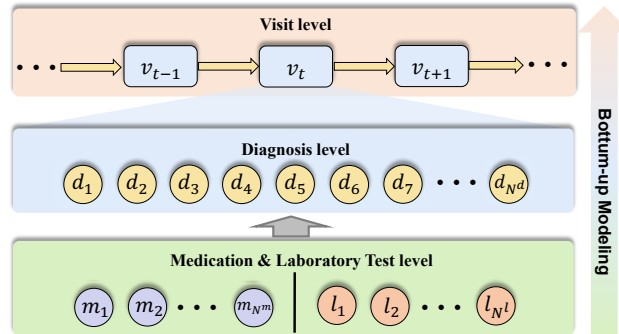

*Figure 3.* The hierarchy for visit-level features.

Finally, the representation for diagnosis codes is derived using the hierarchical architecture described above.

$$\mathcal{R}_{\mathcal{D}_v} = f_{\mathcal{D}}(g_p(\mathcal{D}_v, \mathcal{T}_{\mathcal{O}}), g_s(\mathcal{D}_v, \mathcal{T}_{\mathcal{O}})) \tag{2}$$

where $\mathcal{D}_v$ is the set of diagnosis codes for visit $v$, and $\mathcal{T}_{\mathcal{O}}$ denotes the medical ontology tree. The functions $g_p$ and $g_s$ respectively model the hierarchical path and semantics. The function $f_{\mathcal{D}}$ is responsible for transforming these representations into diagnosis-level representations. Further details regarding the implementations of these functions are provided in Appendix G.

In addition to the hierarchical ontology structure for diagnosis codes, a further code-level hierarchy is discernible within the patients' visits. During a hospital visit, diagnosis codes represent diseases, medications are prescribed to treat these diseases, and laboratory tests assist in confirming the diagnoses. Correspondingly, akin to the approach adopted in MiME (Choi et al., 2018), we establish a code-level hierarchy for visit-level representation, as illustrated in Figure 3. The visit-level representation is computed via:

$$\mathcal{R}_v = f_{\mathcal{V}}(\mathcal{R}_{\mathcal{D}_v}, f_{\mathcal{ML}}(\mathcal{R}_{\mathcal{M}_v}, \mathcal{R}_{\mathcal{L}_v})) \tag{3}$$

where $\mathcal{R}_{\mathcal{D}_v}$ is the representation of diagnosis codes calculated by Equation 2. $\mathcal{R}_{\mathcal{M}_v}$ and $\mathcal{R}_{\mathcal{L}_v}$ denote the representations of medication codes, and laboratory test codes for visit $v$, where $\mathcal{R}_{\mathcal{M}_v} = f_{\mathcal{M}}(\mathcal{M}_v)$, $\mathcal{R}_{\mathcal{L}_v} = f_{\mathcal{L}}(\mathcal{L}_v)$. The function $f_{\mathcal{V}}$ is employed to generate the visit-level representation $\mathcal{R}_v$.

To derive patient-level representations, we take the current visit as the anchor visit and adopt an adapted reverse time attention mechanism.

$$\alpha_{N^v}, \cdots, \alpha_1 = \text{softmax}(\text{GRU}(\mathcal{R}_{v_{N^v}}, \cdots, \mathcal{R}_{v_1})) \tag{4}$$

$$\beta_j = \text{sim}(\mathcal{R}_{v_j}, \mathcal{R}_{v_{N^v}}), \text{for } j = 1, 2, \cdots, N^v \tag{5}$$

$$\mathcal{R}_p = \sum_{i=1}^{N^v} \alpha_v \beta_v \odot \mathcal{R}_{v_i} \tag{6}$$

where the attention $\alpha$ for each visit is calculated in reverse order. We introduce a cosine similarity score $\beta$ to explicitly

model the similarity between past visits and the most recent visit. The combination of $\alpha$ and $\beta$ enhances the model's ability to capture and utilize temporal relationships within the visit sequence.

**PseudoSim Training.** The central consideration for cohort construction is patient similarity. Traditionally, patient similarity is labeled manually by experts, which is a resource-intensive process demanding substantial human efforts. To address this challenge, we propose a pseudo patient similarity prediction task, leveraging pseudo labels generated from patients' diagnosis codes to facilitate the learning of informative patient representations , grounded on the fact that diagnosis codes play a pivotal role as the primary criteria in medical treatment.

Given the temporal context and diagnosis code frequency, we utilize a weighted Jaccard Similarity function enhanced with a time decay factor to determine patient similarity:

$$S(p_i, p_j) = \frac{\sum\limits_{d \in \mathcal{D}_{p_i} \cap \mathcal{D}_{p_j}} e^{-(\Delta t_i + \Delta t_j)} CF_{\min}(d, \mathcal{D}_{p_i}, \mathcal{D}_{p_j})}{\sum\limits_{d \in \mathcal{D}_{p_i} \cup \mathcal{D}_{p_j}} CF_{\max}(d, \mathcal{D}_{p_i}, \mathcal{D}_{p_j})} \quad (7)$$

where $\mathcal{D}_{p_i}$ and $\mathcal{D}_{p_j}$ are the sets of diagnosis codes across all visits for patient $p_i$ and $p_j$, respectively. $\Delta t_i$ and $\Delta t_j$ represent the time intervals between the last occurrence of the diagnosis code $d$ and the most recent visit for patients $p_i$ and $p_j$, respectively. $CF_{\min}$ ($CF_{\max}$) computes the minimum (maximum) frequency of the diagnosis code $d$ in $D_{p_i}$ and $D_{p_j}$.

Finally, we optimize the representation learning by learning patient relationships through maximization of Mutual Information Neural Estimation (Belghazi et al., 2018). For each patient $p$, we select the top $K_p$ patients with the highest similarity scores as positive samples $S_p^+$, and randomly choose $K_p$ patients from the remaining pool as negative samples $S_p^-$. During the training process, we utilize the learned patient-level representations $\mathcal{R}_p$, positive samples $S_p^+$ and negative samples $S_p^-$ for PseudoSim Training.

$$L_{pcs} = -\mathbb{E}_{i \in \mathcal{S}_p^+} f_\phi(\mathcal{R}_p, \mathcal{R}_i) + \log \mathbb{E}_{i \in \mathcal{S}_p^+} \exp(f_\phi(\mathcal{R}_p, \mathcal{R}_i))$$
$$+ \mathbb{E}_{j \in \mathcal{S}_p^-} f_\phi(\mathcal{R}_p, \mathcal{R}_j) - \log \mathbb{E}_{j \in \mathcal{S}_p^-} \exp(f_\phi(\mathcal{R}_p, \mathcal{R}_j)) \quad (8)$$

where the first line represents the mutual information of positive samples composed of expectation of similarity and log-normalization, and the second line serves as a penalty for the negative samples. $f_\phi$ is the similarity prediction network.

The strategy above not only yields valuable information on patient similarity but also facilitates the construction of cohorts $\mathcal{C}$ from the patient-level representations $\mathcal{R}_p$, which is subsequently harnessed in downstream tasks.

**Cohort Derivation.** The patient-level representations could potentially be utilized with various methods for cohort derivation; however, it is beyond the primary scope of this paper. Here, Jensen-Shannon divergence (Lin, 1991) and student's t-distribution (Student, 1908) are employed. Please refer to Appendix G for detailed implementation.

$$L_{JS} = \sum_p \sum_c t_{pc} \log \frac{2t_{pc}}{t_{pc} + q_{pc}} + q_{pc} \log \frac{2q_{pc}}{q_{pc} + t_{pc}} \quad (9)$$

where $q_{pc} = \frac{(1 + \|\mathcal{R}_p - \mu_c\|^2)^{-1}}{\sum_i^{|C|}(1 + \|\mathcal{R}_p - \mu_i\|^2)^{-1}}$, $t_{pc} = \frac{q_{pc}^2 / \sum_i q_{ic}}{\sum_j^{|C|}(q_{pj}^2 / \sum_i q_{ij})}$. $q_{pc}$ is the soft cohort assignment of patient $p$, $t_{pc}$ is the self-training target cohort distribution, $|C|$ is the number of cohorts and $\mu_c$ is the centroid of cohort $c$. Finally, the cohort $c$ for patient $p$ is $\arg\max\limits_c q_{pc}$.

### 3.3. Biscale Cohort Learning Module

Cohorts play a crucial role in EHR analysis. The Biscale Cohort Learning Module is employed to encode cohort information into the representations derived from backbones and harness local and global cohort information to its fullest extent for diverse healthcare analytic tasks.

**Individual Representation Initialization.** We first derive individual representations from EHR data using established backbones as the initial representations. For a given patient $p$, the initial representation $\mathcal{R}_0^p$ is generated by the backbones, which will serve as fundamental inputs for NeuralCohort, facilitating its subsequent operations.

**Local Intra-cohort Modeling.** To effectively capture interactions among patients within a cohort, Local Intra-cohort Modeling depicts the cohort as a graph in which each patient is represented as a node. In this graph, each pair of patients in the same cohort is connected. The weight between two nodes is calculated based on the similarity between patient representations $\mathcal{R}_p$ computed in the Pre-context Cohort Synthesis Module.

$$\mathcal{A}(i, j) = \frac{\mathcal{R}_i^T \mathcal{R}_j}{\|\mathcal{R}_i\| \|\mathcal{R}_j\|} \quad (10)$$

$$\mathcal{H}^{k+1} = \sigma(\hat{\mathcal{A}} \mathcal{H}^k \mathcal{W}^k + b^k) \quad (11)$$

where $\mathcal{A}$ is the adjacency matrix of the graph. $\mathcal{H}^k$ is the node representations at layer $k$, $\mathcal{H}^0 = \mathcal{R}_0$. $\mathcal{W}^k$ and $b^k$ are the trainable weights and bias. $\hat{\mathcal{A}} = \hat{D}^{-\frac{1}{2}}(\mathcal{A} + \mathcal{I})\hat{D}^{-\frac{1}{2}}$ is the normalized adjacency matrix and $\hat{D}$ is the degree matrix. The output of the final layer is the local intra-cohort representation $\mathcal{R}_L$.

**Global Inter-cohort Modeling.** The global inter-cohort modeling aims to elucidate the contrasting knowledge across dissimilar cohorts. The inter-cohort representations should maintain distinct separation while preserving the semantic integrity of each cluster. In light of this, we employ an encoder-decoder architecture and cohort loss function to reconstruct the representations for cohorts.

$$\mathcal{R}_G^c = f_{dec}(f_{enc}(\frac{1}{|c|} \sum_{p \in c} \mathcal{R}_0^p)) \quad (12)$$

Patients within the same cohort generally exhibit similar behaviors, in contrast to those from different cohorts, who may manifest distinct patterns. To supervise the representations of both intra- and inter-cohort graphs, we introduce a cohort loss function composed of reconstruction loss and contrastive loss:

$$L_{co} = \sum_{c \in C} ||\mathcal{R}_G^c - \frac{1}{|c|}\sum_{p \in c}\mathcal{R}_0^p||_F^2 - \frac{\sum\limits_{c \in \mathcal{C}} \sum\limits_{i,j \in c} sim(\mathcal{R}_L^i, \mathcal{R}_L^j)}{\sum\limits_{c,c' \in \mathcal{C}} sim(\mathcal{R}_G^c, \mathcal{R}_G^{c'})} \quad (13)$$

where sim is the cosine similarity function. The first term is the reconstruction loss and the other term is the cohort contrastive loss.

**Cohort Fusion.** To consolidate the information from the representations generated by the backbones, the intra-cohort graph, and the inter-cohort graph, we devise a cross-domain attention mechanism to obtain the augmented EHR data representation for subsequent healthcare analytic tasks:

$$\alpha_L = \frac{R_0^p \cdot R_L^p}{\sqrt{\dim(R_0^p)}}, \alpha_G = \frac{R_0^p \cdot R_G^c}{\sqrt{\dim(R_0^p)}} \quad (14)$$

$$R_{final}^p = f_{attn}(R_0^p, \alpha_L \cdot R_L^p, \alpha_G \cdot R_G^c) \quad (15)$$

where $R_L^p$ and $R_G^{C^p}$ pertain to the representations extracted from the intra- and inter-cohort graphs, respectively. The final loss function for NeuralCohort is as follows:

$$L = \lambda_{pcs}L_{pcs} + \lambda_{JS}L_{JS} + \lambda_{co}L_{co} + L_{ds} \quad (16)$$

where $\lambda_{pcs}$, $\lambda_{JS}$ and $\lambda_{co}$ denote the weights assigned to the loss of the PseudoSim Training $L_{pcs}$, cohort derivation $L_{JS}$ and the Cohort Loss $L_{co}$. $L_{ds}$ signifies the loss associated with the downstream task.

# 4. Experiments

## 4.1. Experimental Setup

We conduct extensive experiments across three widely-recognized real-world EHR datasets: MIMIC-III (Johnson et al., 2016), MIMIC-IV (Johnson et al., 2020) and Diabetes130 (Clore John & Beata, 2014), addressing two pivotal medical tasks, cross-visit Hospital Readmission Prediction (Huang et al., 2019) and within-visit Long Length-of-Stay (LOS) Prediction (Alsinglawi et al., 2022). Through these experiments, we aim to assess the generalizability and efficacy of NeuralCohort in predicting critical healthcare outcomes, thereby demonstrating its applicability across diverse patient populations and clinical scenarios.

To assess the versatility of NeuralCohort as a plug-in for enhancing the representations derived from various backbones, we compare the performance of baselines with their enhanced counterparts using cohort information provided by NeuralCohort. We employ well-established backbone models, namely Med2Vec (Choi et al., 2016), MiME (Choi

Table 1. Overall performance of NeuralCohort against baselines for readmission prediction on the MIMIC-III dataset.

| Model | Readmission Task on MIMIC-III | | |
| --- | --- | --- | --- |
| | **AUPRC** | **AUROC** | **Accuracy** |
| ClinicalBERT | 0.630±0.005 | 0.651±0.006 | 58.7%±0.5% |
| + KNN | 0.628±0.002 | 0.651±0.002 | 58.6%±0.2% |
| + K-Means | 0.629±0.001 | 0.650±0.001 | 58.4%±0.3% |
| + DEC | 0.632±0.005 | 0.654±0.002 | 58.4%±0.1% |
| + DEKM | 0.638±0.003 | 0.659±0.005 | 58.9%±0.2% |
| + GRASP | 0.618±0.002 | 0.617±0.001 | 56.2%±0.1% |
| + DGLoS | 0.635±0.003 | 0.533±0.002 | 58.0%±0.4% |
| + IDC | 0.638±0.003 | 0.657±0.004 | 59.0%±0.3% |
| **+ NeuralCohort** | **0.662±0.003** | **0.681±0.005** | **61.2%±0.4%** |
| Med2Vec | 0.554±0.005 | 0.614±0.004 | 54.1%±0.7% |
| + KNN | 0.541±0.004 | 0.598±0.003 | 54.9%±0.5% |
| + K-Means | 0.544±0.005 | 0.600±0.004 | 54.5%±0.4% |
| + DEC | 0.550±0.003 | 0.611±0.002 | 54.3%±0.5% |
| + DEKM | 0.547±0.004 | 0.608±0.006 | 54.6%±0.4% |
| + GRASP | 0.542±0.003 | 0.601±0.006 | 53.8%±0.3% |
| + DGLoS | 0.559±0.004 | 0.542±0.002 | 54.5%±0.5% |
| + IDC | 0.562±0.004 | 0.622±0.003 | 54.5%±0.4% |
| **+ NeuralCohort** | **0.574±0.003** | **0.634±0.005** | **56.9%±0.2%** |
| MiME | 0.543±0.006 | 0.602±0.005 | 56.8%±0.5% |
| + KNN | 0.543±0.004 | 0.610±0.003 | 56.5%±0.5% |
| + K-Means | 0.546±0.004 | 0.605±0.006 | 56.5%±0.7% |
| + DEC | 0.549±0.007 | 0.608±0.004 | 57.3%±0.9% |
| + DEKM | 0.548±0.002 | 0.611±0.003 | 57.1%±0.5% |
| + GRASP | 0.530±0.009 | 0.589±0.010 | 57.2%±0.9% |
| + DGLoS | 0.551±0.006 | 0.543±0.004 | 57.6%±0.7% |
| + IDC | 0.542±0.006 | 0.605±0.003 | 57.2%±0.4% |
| **+ NeuralCohort** | **0.568±0.004** | **0.629±0.003** | **58.6%±0.3%** |

et al., 2018), and ClinicalBERT (Huang et al., 2019) to derive initial representation $\mathcal{R}_0$. Subsequently, we employ KNN, K-Means, DEC (Xie et al., 2016), DEKM (Guo et al., 2021), GRASP (Zhang et al., 2021), DGLoS (Zang et al., 2023) and IDC (Svirsky & Lindenbaum, 2024) as baselines for comparison with our proposed NeuralCohort, which are directly applied to the representations of the backbones to integrate the cohort information into $\mathcal{R}_0$ for enhanced predictive accuracy. The efficacy of the models on a given task primarily hinges on the backbones. The objective of NeuralCohort is to infuse patient similarity information into these backbones. Therefore, our analysis focuses on assessing the performance enhancements achieved by integrating NeuralCohort and other baselines into the backbones.

In our experimental setup, we repeat each model across the datasets five times and compute the average result. We measure the performance of the models using three metrics: Area Under the Precision-Recall Curve (AUPRC), Area Under the Receiver Operating Characteristic (AUROC), and accuracy. Additionally, we conduct an ablation study of NeuralCohort and a sensitivity study on its key hyperparameters including $K_p$ and the number of cohorts $|C|$.

Detailed descriptions of the datasets, tasks, backbones, baselines, and experimental settings are provided in Appendix from D to G.

*Table 2.* Overall performance of NeuralCohort against baselines for long LOS prediction on the `MIMIC-III` dataset.

| Model | Long LOS Task on `MIMIC-III` | | |
|---|---|---|---|
| | **AUPRC** | **AUROC** | **Accuracy** |
| ClinicalBERT | 0.658±0.002 | 0.590±0.002 | 59.7%±0.3% |
| + KNN | 0.659±0.002 | 0.594±0.001 | 59.9%±0.4% |
| + K-Means | 0.655±0.001 | 0.590±0.002 | 59.8%±0.3% |
| + DEC | 0.652±0.008 | 0.603±0.009 | 60.2%±0.3% |
| + DEKM | 0.661±0.003 | 0.592±0.002 | 60.1%±0.3% |
| + GRASP | 0.665±0.002 | 0.584±0.003 | 60.0%±0.4% |
| + DGLoS | 0.685±0.005 | 0.548±0.004 | 61.5%±0.9% |
| + IDC | 0.681±0.004 | 0.612±0.003 | 60.2%±0.4% |
| **+ NeuralCohort** | **0.738±0.003** | **0.671±0.004** | **63.7%±0.7%** |
| Med2Vec | 0.908±0.001 | 0.894±0.001 | 64.4%±0.3% |
| + KNN | 0.876±0.002 | 0.876±0.001 | 75.1%±0.4% |
| + K-Means | 0.877±0.002 | 0.875±0.003 | 74.8%±0.3% |
| + DEC | 0.886±0.004 | 0.889±0.006 | 75.4%±0.7% |
| + DEKM | 0.903±0.002 | 0.897±0.004 | 73.8%±0.4% |
| + GRASP | 0.903±0.003 | 0.887±0.002 | 69.0%±0.5% |
| + DGLoS | 0.907±0.004 | 0.837±0.006 | 75.9%±0.5% |
| + IDC | 0.906±0.003 | 0.899±0.003 | 73.2%±0.4% |
| **+ NeuralCohort** | **0.919±0.002** | **0.906±0.004** | **80.7%±0.3%** |
| MiME | 0.913±0.003 | 0.904±0.003 | 78.4%±0.5% |
| + KNN | 0.910±0.002 | 0.895±0.003 | 80.2%±0.4% |
| + K-Means | 0.904±0.003 | 0.891±0.002 | 79.0%±0.3% |
| + DEC | 0.912±0.005 | 0.907±0.008 | 79.6%±0.4% |
| + DEKM | 0.911±0.005 | 0.908±0.007 | 79.6%±0.4% |
| + GRASP | 0.898±0.004 | 0.896±0.003 | 81.1%±0.4% |
| + DGLoS | 0.917±0.002 | 0.854±0.008 | 80.7%±0.3% |
| + IDC | 0.919±0.005 | 0.910±0.004 | 80.3%±0.2% |
| **+ NeuralCohort** | **0.936±0.004** | **0.923±0.002** | **82.8%±0.2%** |

We provide additional experimental evaluations, including a sensitivity study in Appendix H and an efficiency study in Appendix J.

### 4.2. Overall Results

We present the overall results of NeuralCohort and baselines for the two prediction tasks on `MIMIC-III` dataset in Tables 1 and 2, respectively. The conclusions drawn from different datasets are consistent, confirming the robustness of our findings. Results for `MIMIC-IV` and `Diabetes130` datasets are provided in Appendix I.

As demonstrated by the results, NeuralCohort consistently attains the best performance, outperforming the baselines by a significant margin with improvements up to 8.0% in AUPRC, 8.1% in AUROC, and 16.3% in accuracy. This considerable performance advantage underscores the efficacy of NeuralCohort over diverse backbones. Further, NeuralCohort enhances the performance of each backbone on both tasks on all evaluated metrics, which substantiates its capability to derive fine-grained cohorts and effectively infuse cohort information into the representations, leading to improved predictive accuracy.

We observe the baselines fail to yield consistent performance improvements primarily due to their inadequate modeling of cohort information at a fine-grained level for accurate

*Table 3.* The ablation study results of NeuralCohort on long LOS prediction utilizing the ClinicalBERT backbone on the `MIMIC-III` dataset. $M_P$, $M_{intra}$, and $M_{inter}$ represent Pre-context Cohort Synthesis Module, Local Intra-cohort Modeling, and Global Inter-cohort Modeling of Biscale Cohort Learning Module.

| Model | Modules | | | `MIMIC-III` | | |
|---|---|---|---|---|---|---|
| | $M_P$ | $M_{intra}$ | $M_{inter}$ | **AUPRC** | **AUROC** | **Accuracy** |
| -$M_P$ | | ✔ | ✔ | 0.682 | 0.613 | 60.4% |
| -$M_{intra}$ | ✔ | | ✔ | 0.675 | 0.603 | 59.2% |
| -$M_{inter}$ | ✔ | ✔ | | 0.714 | 0.649 | 61.8% |
| **NeuralCohort** | ✔ | ✔ | ✔ | **0.738** | **0.671** | **63.7%** |

analysis. Specifically, KNN and K-Means, when applied directly to backbones, do not operate within a similarity-aware feature space. Additionally, DGLoS constructs a coarse-grained global graph, and GRASP focuses exclusively on inter-cohort modeling. DEC, DEKM and IDC cannot model the medical semantics. As a result, instead of accurately modeling patient similarity, these baselines may introduce noise regarding patient similarity into the backbones, consequently degrading the overall performance.

Among the backbones, ClinicalBERT demonstrates superior performance compared to Med2Vec and MiME in the hospital readmission task, whereas the opposite trend is observed in the long LOS task. This divergence can be attributed to the distinct characteristics of the tasks and the underlying model architectures. Med2Vec and MiME are adept at modeling relationships among a large number of medical codes, which provide comprehensive information about a patient's current visit, thus making them more suitable for within-visit long LOS prediction. Conversely, clinical notes contain extensive records of patients' past medical histories. ClinicalBERT, which excels at learning representations from such data, is better equipped to capture the relationships across multiple visits and enhance its efficacy for cross-visit readmission prediction.

### 4.3. Ablation Study

We conduct an ablation study on the long LOS prediction task with the ClinicalBERT backbone on the `MIMIC-III` dataset, in order to evaluate the impact of each module and component of NeuralCohort. The experimental results are presented in Table 3. Comparing NeuralCohort against NeuralCohort (-$M_P$) (Row 1 vs. 4), the Pre-context Cohort Synthesis Module demonstrates notable contributions, *i.e.*, 5.6%, 5.8%, and 3.3% improvements in AUPRC, AUROC, and accuracy. This underscores the efficacy of the Pre-context Cohort Synthesis Module in capturing nuanced patient similarity, thus emphasizing the importance of accounting for fine-grained cohorts for EHR Representation Learning. The results in Rows 2 and 3 of Table 3 elucidate

*Table 4.* Comparison between traditional cohorts and NeuralCohort on the backbone ClinicalBERT and Med2Vec for readmission prediction on the `MIMIC-III` dataset.

| Model | MIMIC-III | | |
|---|---|---|---|
| | **AUPRC** | **AUROC** | **Accuracy** |
| ClinicalBERT | 0.630 | 0.651 | 58.7% |
| + $MC_G$ | 0.629 | 0.651 | 58.5% |
| + $MC_A$ | 0.631 | 0.651 | 58.3% |
| + $MC_D$ | 0.629 | 0.652 | 58.7% |
| + $MC_H$ | 0.621 | 0.643 | 58.1% |
| + **NeuralCohort** | **0.662** | **0.681** | **61.2%** |
| Med2Vec | 0.554 | 0.614 | 54.1% |
| + $MC_G$ | 0.546 | 0.611 | 54.3% |
| + $MC_A$ | 0.548 | 0.612 | 53.4% |
| + $MC_D$ | 0.557 | 0.619 | 54.5% |
| + $MC_H$ | 0.551 | 0.616 | 53.7% |
| + **NeuralCohort** | **0.574** | **0.634** | **56.9%** |

the respective contributions of Local Intra-cohort Modeling and Global Inter-cohort Modeling. In comparison, the removal of the Local Intra-cohort Modeling results in a more pronounced decline in performance compared to the removal of Global Inter-cohort Modeling. This highlights the necessity of unveiling implicit information within similar patients. In a nutshell, these results affirm the indispensable roles played by distinct modules and components in NeuralCohort in ensuring accurate EHR analysis.

### 4.4. Traditional Cohorts *vs.* NeuralCohort

To evaluate NeuralCohort against traditional medical cohort construction approaches in effectiveness, we employ a widely recognized cohort division approach using two distinct criteria: gender (G), age (A) and two disease-specific criteria: diabetes diagnosis (D) and hypertension diagnosis (H). The results are displayed in Table 4. As shown, traditional medical cohorts tend to perform comparably to, and occasionally less accurately than the backbones. This primarily arises from the reliance of traditional medical cohort construction on a limited set of features, resulting in coarse-grained cohorts that are not effective enough for cohort pattern mining. Consequently, such traditional approaches will inevitably group highly dissimilar patients into the same cohort, introducing noise to the model that adversely affects the overall performance. In stark contrast, NeuralCohort leverages a patient's sequential visit-level representations from both intra- and inter-cohort perspectives at a fine-grained level, thereby boosting the prediction accuracy.

### 4.5. Interpretability Analysis

**Quantitive Analysis of Cohorts.** It is non-trivial to define patient similarity for comparison in healthcare analytics,

*Table 5.* Comparison of C-H scores between NeuralCohort and baselines with the backbones ClinicalBERT and Med2Vec on long LOS prediction.

| Model | $S_{C-H}$ | Model | $S_{C-H}$ |
|---|---|---|---|
| ClinicalBERT | 57.6 | Med2Vec | 18.4 |
| + KNN | 56.8 | + KNN | 18.5 |
| + K-Means | 59.7 | + K-Means | 20.1 |
| + DEC | 62.6 | + DEC | 20.9 |
| + DEKM | 60.2 | + DEKM | 19.6 |
| + GRASP | 65.5 | + GRASP | 21.4 |
| + DGLoS | 61.7 | + DGLoS | 21.5 |
| + IDC | 69.4 | + IDC | 23.2 |
| + **NeuralCohort** | **80.3** | + **NeuralCohort** | **25.7** |

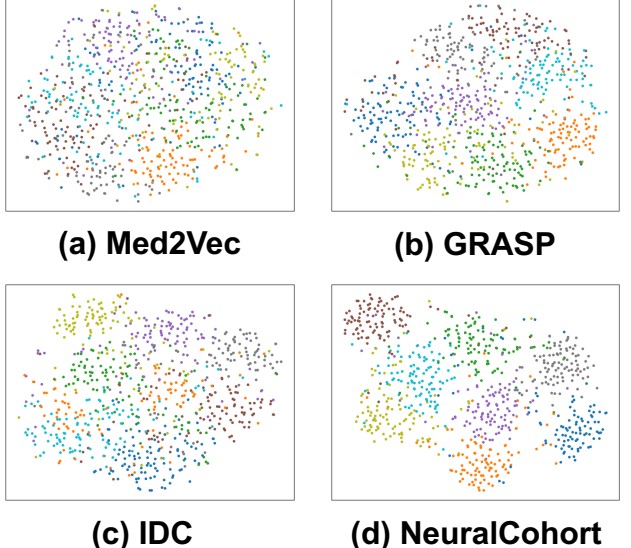

**(a) Med2Vec**      **(b) GRASP**

**(c) IDC**      **(d) NeuralCohort**

*Figure 4.* t-SNE visualization of the final representations before prediction for selected eight-cohort points, where backbone is Med2Vec.

which further results in the absence of ground truth for evaluating the performance of clustering-based methods on patient similarity. To address this, we use the Calinski-Harabasz score (C-H score) (Caliński & Harabasz, 1974) to quantitatively assess the performance of NeuralCohort and other baselines. The C-H score $S_{C-H}$ is calculated as:

$$S_{C-H} = \frac{\text{tr}(B)*(m-|\mathcal{C}|)}{\text{tr}(W)*(|\mathcal{C}|-1)} \quad (17)$$

where $m$ is the sample size, $|\mathcal{C}|$ is the number of clusters, $B$ is the cross-cluster variance, $W$ is the within-cluster variance, and $tr$ is the trace of matrix.

The comparison of C-H scores between NeuralCohort and baselines with ClinicalBERT and Med2Vec for the long LOS prediction task is detailed in Table 5. We note that KNN, DGLoS, along with the bare backbones do not involve the concept of "clusters" and as a result, we employ K-

Means to their representations prior to prediction layer to evaluate C-H scores. The results clearly demonstrate that integrating a backbone with NeuralCohort yields higher C-H scores, which indicates that the cohorts generated through NeuralCohort are more interpretable and effective.

**Visualization.** The visualization of selected eight cohorts (clusters) points using t-SNE (Van der Maaten & Hinton, 2008), based on the final representation $R^p_{final}$ of Med2Vec for the long LOS prediction task, is illustrated in Figure 4. Extended visualization of all baselines are shown in Appendix K. Notably, significant overlaps among the points from Figure 4 (a) to (c) suggest that the backbone and baselines lack robust discriminative capabilities when utilizing $R^p_{final}$. However, the integration of NeuralCohort substantially reduces this overlap, enhancing the distinctness of the cohorts. This demonstrates that NeuralCohort is capable of encoding the data into a representation similarity space and the fine-grained cohort information integrated by NeuralCohort effectively facilitates downstream EHR analysis tasks.

### 4.6. Clinical Significance

NeuralCohort identifies cohort-specific features that are directly linked to clinical outcomes, thereby enhancing decision-making for patient management. For example, the distinctive features of the four cohorts visualized in Figure 4 are provided in Table 6, where the features are identified through t-test. The detailed descriptions of the used abbreviations are listed in Table 7. Specifically, *Cohort #1* includes patients with cardiovascular conditions and is at high risk for acute cardiac events and extended hospital stays (Tigabe Tekle et al., 2022). Early identification enables the prioritization of telemetry beds, cardiology consults, and monitoring. Clinically, this enables timely diuretics, echocardiography, and discharge planning with heart failure follow-up. *Cohort #2* features patients with chronic metabolic and hematologic conditions that often co-occur and need interdisciplinary care. Identifying this cohort allows hospitals to mobilize diabetes educators, lipid clinics, and anticoagulation monitoring services. Clinically, it enables insulin titration, lipid therapy, and bleeding risk management to reduce complications and improve outcomes. *Cohort #3* is characterized by renal and urological issues requiring frequent labs, fluid monitoring, and nephrology support. Early identification allows hospitals to allocate renal panels, schedule imaging for uropathy, and prepare dialysis resources—helping manage AKI risk and avoid care escalation. *Cohort #4* presents complex chronic and acute conditions, requiring coordination across pulmonology, nephrology, endocrinology, and infectious disease. Identifying this cohort supports planning for respiratory support, hormone therapy, infection control, and opioid withdrawal. Operationally, it enables resource allocation for respiratory teams,

*Table 6.* Distinctive feature of four cohorts

| Cohort | Distinctive features |
|---|---|
| #1 | HF, Arrhythmia, CHD, Angina |
| #2 | Diabetes, DLM, CD, Obesity |
| #3 | OUA, PSL, UF, F&E Disorders |
| #4 | OPI, CKD, Thyroiditis, Flu |

*Table 7.* Detailed Descriptions of the Abbreviations

| Abbreviation | Detailed Description |
|---|---|
| HF | Heart Failure |
| CHD | Chronic Heart Disease |
| DLM | Disorders of Lipoid Metabolism |
| CD | Coagulation Defects |
| OUA | Other and Unspecified Anemias |
| PSL | Pneumonitis due to Solids and Liquids |
| UF | Urethral Fistula |
| F & E Disorders | Fluid and Electrolyte Disorders |
| OPI | Other Pulmonary Insufficiency |
| CKD | Chronic Kidney Disease |

isolation rooms, and endocrine/renal labs. Such insights allow clinicians to optimize resource allocation and tailor interventions, thus significantly improving hospital efficiency and patient care.

## 5. Conclusions

In EHR representation learning, existing studies primarily focus on deriving informative representations to facilitate downstream task predictions. However, two essential desiderata, fine-grained cohort division and effective exploitation of both intra- and inter-cohort information, remain unresolved. To address the problem, we propose NeuralCohort, a cohort-aware neural representation learning method for EHR representation learning. The core thrust of NeuralCohort centers around two pivotal modules: the Pre-context Cohort Synthesis Module, responsible for cohort derivation, and the Biscale Cohort Learning Module, dedicated to capturing and encoding local intra-cohort and global inter-cohort insights. Extensive experimental results confirm the superiority of NeuralCohort over baselines across various backbones, achieving improvements of up to 8.1% in AU-ROC. More importantly, NeuralCohort contributes valuable medical insights to EHR analysis.

## Acknowledgments

The work of NUS researchers is supported by the Lee Foundation in terms of NUS Lee Kong Chian Centennial Professorship grant.

## Impact Statement

This paper advances deep learning for healthcare by addressing critical challenges in cohort discovery. The proposed NeuralCohort extends beyond traditional individual intelligence paradigms, evolving into interpretable cohort intelligence to enhance patient population analysis. Being highly adaptable and generalizable, NeuralCohort can be seamlessly integrated with diverse backbone models for a wide variety of healthcare applications. More importantly, Neural-Cohort holds significant implications for large-scale cohort research, fostering new insights into population health dynamics, supporting advancements of precision medicine, and ultimately contributing to the development of more effective and targeted healthcare interventions.

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

## A. Notations

To ensure clarity and consistency throughout the paper, a notation table has been provided. This table serves as a quick reference for readers to easily understand and recall the meaning of each symbol.

*Table 8.* Frequently used notations

| Notation | Meaning |
|---|---|
| $\mathcal{P}$ | The set of all the patients |
| $p_i$ | $i$-th patient in the set of patient $P$ |
| $N^p$ | Number of patients |
| $\mathcal{V}_p$ | Set of visits for patient $p$ |
| $v_i$ | $i$-th visit in $\mathcal{V}_p$ |
| $N^v$ | Number of visits for patient $p$ |
| $d_i$ | $i$-th diagnosis code within a visit similar to $m_i$ and $l_i$ |
| $N^d$ | Number of diagnosis codes for visit $v$ similar to $N^m$ and $N^l$ |
| $\mathcal{D}_v$ | Diagnosis codes of visit $v$ similar to $\mathcal{M}_v$ and $\mathcal{L}_v$ |
| $\mathcal{D}_{p_i}$ | the sets of diagnosis codes across all visits for patient $p_i$ |
| $f_{\mathcal{V}}$ | network to generate the visit-level representation similar to other functions $f_{\mathcal{M}}, f_{\mathcal{L}}, f_{\mathcal{D}}$ and $f_{\mathcal{ML}}$ |
| $f_\phi$ | the similarity prediction network in PseudoSim Training |
| $f_{enc}$ | the encoder used in Global Inter-cohort Modeling |
| $f_{dec}$ | the decoder used in Global Inter-cohort Modeling |
| $\mathcal{R}_{\mathcal{D}_v}$ | Representation of diagnosis codes of visit $v$ similar to $\mathcal{R}_{\mathcal{M}_v}$ and $\mathcal{R}_{\mathcal{L}_v}$ |
| $\mathcal{R}_v$ | Representation of visit $v$ |
| $\mathcal{R}_p$ | Representation of patient $p$ based on $\mathcal{R}_v$ |
| $S(p_i, p_j)$ | pseudo similarity between patient $p_i$ and $p_j$ |
| $S_p^+$ | Positive patients for patient $p_i$ based on their similarity |
| $S_p^-$ | Negative patients for patient $p_i$ based on their similarity |
| $\mathcal{T}_{\mathcal{O}}$ | Medical ontology tree |
| $\mathcal{C}$ | Generated cohorts |
| $\mathcal{R}_0^p$ | Initial representation of patient $p$ derived from backbone |
| $\mathcal{R}_L^p$ | representation of patient $p$ in Local Intra-cohort Modeling |
| $\mathcal{R}_G^c$ | representation of cohort $c$ in Global Inter-cohort Modeling |
| $K_p$ | Number of patients for $S_p^+$ or $S_p^-$ |
| $\alpha_L$ | the attention for Local Intra-cohort Modeling |
| $\alpha_G$ | the attention for Global Inter-cohort Modeling |

## B. Extended Related Work

**Traditional Cohort Study**. Cohorts have been broadly applied across various practical domains to analyze group-based patterns and outcomes (Xiao et al., 2024), with the help of cohort processing techniques (Jiang et al., 2016; Xie et al., 2018; Cai et al., 2018; Xie et al., 2020). Among these domains, healthcare stands out as an especially critical area where cohort analysis plays a central role. Cohort studies in healthcare are a type of observational research design that longitudinally tracks individuals sharing specific characteristics over time (Eldredge, 2002; Setia, 2016). The primary objective of these studies is to investigate the emergence of particular health-related outcomes or events. Cohort studies hold substantial significance in healthcare research, as they facilitate the examination of relationships

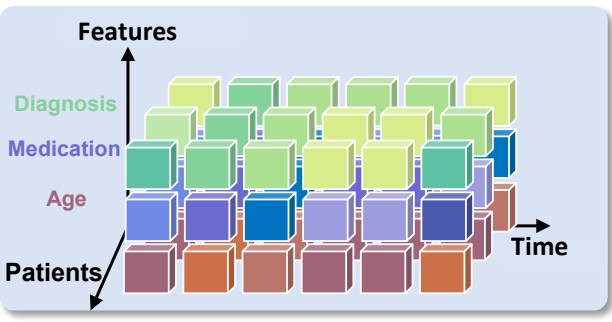

*Figure 5.* Complex EHR data structure from three dimensions. EHR data consists of time-series medical information that varies across patients.

between potential risk factors or interventions and various health outcomes (Paneth & Monk, 2018; Song & Chung, 2010; Cai et al., 2024; Zheng et al., 2024). For example, a substantial and demographically diverse cohort of HIV-infected treatment-naïve participants is leveraged to explore correlations between viral load and amino acid entropy, considering variables such as sex, age, race, duration of infection, and HIV population structure (Gabrielaite et al., 2023). However, such traditional cohort studies in healthcare generally employ a coarse-grained approach, leading to an insufficient exploration of nuanced patient insights.

**Deep Clustering**. Clustering is a fundamental technique to group similar data points together based on their intrinsic structure. The clusters provide a comprehensive global characterization of instances, which could be used for interpretable feature learning (Svirsky & Lindenbaum, 2024), dimensionality reduction (Ros & Riad, 2024), anomaly detection (Zhu et al., 2022; 2023) and community detection (Li et al., 2017; Wu et al., 2022). With the rapid development and remarkable success of deep learning, deep clustering has emerged as a powerful paradigm to enhance clustering performance through deep neural networks. DEC (Xie et al., 2016) refines cluster assignments using an autoencoder with soft assignment. DIVIDE (Lu et al., 2024) employs high-order random walks to progressively identify reliable data pairs. Despite these advancements, interpretability remains a significant challenge in deep clustering. Unlike traditional clustering approaches such as K-Means and hierarchical clustering, where cluster assignment rules are explicitly defined, deep clustering methods typically operate within high-dimensional latent spaces, making it difficult to interpret how individual samples are grouped. To address this issue, X-DC (Watanabe & Kameoka, 2021) interprets deep clustering as fitting learnable spectrogram templates to an input spectrogram followed by Wiener filtering. IDC (Svirsky & Lindenbaum, 2024) introduce a self-supervised approach to enable sample-specific feature selection, thereby refining cluster assignments. Interpretability is of vital importance in critical domains such as healthcare, finance, and sci-entific discovery. As deep clustering continues to evolve, addressing interpretability concerns will be crucial for their practical adoption in real-world applications.

## C. EHR data

The introduction of EHR data structure is illustrated in Figure 5. EHR data encapsulates patients' medical histories, providing a holistic view of their healthcare journey. As a centralized and structured repository, EHR data integrates various aspects of patient care, including detailed records of hospital admissions, outpatient visits, and emergency department encounters. EHR data comprises a diverse set of information, such as clinical diagnoses, prescribed treatments, medication history, laboratory test results, imaging reports, and clinical notes. Furthermore, EHR systems facilitate longitudinal tracking of patient health, enabling clinicians to monitor disease progression, assess treatment efficacy, and support clinical decision-making. EHR data serve as a critical resource for personalized medicine, healthcare analytics, and large-scale population health studies.

## D. Dataset Descriptions and Statistics

**MIMIC-III** (Johnson et al., 2016) is a publicly available medical database, readily available to researchers worldwide. `MIMIC-III` encompasses 53,423 distinct hospital admissions, catering to adult patients aged 16 years or above, who were admitted to critical care units in the Beth Israel Deaconess Medical Center between 2001 and 2012. Each admission comprises a varying number of vital signs, diagnosis codes, medication codes, and laboratory test codes. Further, there are 2,083,180 de-identified notes linked to these admissions.

**MIMIC-IV** (Johnson et al., 2020) is an openly accessible medical database comprising patient admission data collected from 2008 to 2022. It includes contemporary data and is structured with modular data organization, with an emphasis on data provenance. Its modular design facilitates the independent use of disparate data sources as well as their combined utilization.

**Diabetes130 Dataset** (Clore John & Beata, 2014) consists of ten years (1999-2008) of clinical care data from 130 U.S. hospitals and integrated delivery networks. The dataset is provided for the pattern analysis in historical diabetes care, with the potential to achieve safe and personalized healthcare for patients.

The detailed statistics for `MIMIC-III`, `MIMIC-IV` and `Diabetes130` datasets, including the numbers of patients, visits, and medical codes are presented in Table 9.

*Table 9.* Data statistics of `MIMIC-III` and `MIMIC-IV`.

| Dataset | MIMIC-III | MIMIC-IV | Diabetes130 |
|---|---|---|---|
| # of $\mathcal{P}$ | 38,597 | 299,713 | 71,518 |
| # of $\mathcal{V}$ | 53,423 | 431,231 | 101,766 |
| # of notes | $\approx 2,000,000$ | $\approx 330,000$ | - |
| Avg. # of $\mathcal{V}$ per patient | 1.38 | 1.44 | 1.42 |
| # of unique $\mathcal{D}$ | 6,984 | 25,809 | 915 |
| # of unique $\mathcal{M}$ | 4,686 | 6,911 | 24 |
| # of unique $\mathcal{L}$ | 726 | 1,623 | 2 |
| Avg. # of $\mathcal{D}$ per visit | 12.2 | 11 | 2.98 |
| Avg. # of $\mathcal{M}$ per visit | 78 | 36 | 1.95 |
| Avg. # of $\mathcal{L}$ per visit | 521 | 274 | 0.22 |

## E. Downstream Tasks

**Hospital Readmission Prediction Task** involves predicting whether a patient will be readmitted to the hospital within a 30-day timeframe per visit (Huang et al., 2019). This cross-visit prediction task could provide healthcare professionals with early insights into patients who are at an elevated risk of being readmitted, allowing for timely interventions and improved patient outcomes.

**Long Length-of-Stay (LOS) Prediction Task** pertains to a classification task that predicts whether the patient will remain hospitalized for a duration exceeding seven days (Alsinglawi et al., 2022). Within-visit long LOS prediction could aid in optimizing resource allocation, improving patient flow, and enhancing patient care planning.

## F. Backbones and Baselines

**Backbones** are employed to derive the initial representations $\mathcal{R}_0^p$. In the experiments, Med2Vec, MiME, and ClinicalBERT are employed as baselines for performance evaluation.

- **Med2Vec** (Choi et al., 2016) not only models the co-occurrence information of medical codes but also creates a hierarchical structure based on the sequential order of visits to learn the representations.

- **MiME** (Choi et al., 2018) constructs a hierarchical architecture that decomposes the representations into treatment level, diagnosis level, visit level, and patient level, in order to render the learned representations interpretable and effective.

- **ClinicalBERT** (Huang et al., 2019) adapts the BERT model for healthcare, focusing solely on medical notes. To handle long sequences of medical notes, ClinicalBERT splits them into fixed-length slices, meaning that the slices within a visit share the same label. The final predictions are computed by aggregating the predictions from each slice.

Each dataset contains diverse types of clinical data, and we tailor data usage to each backbone. Med2Vec and MiME primarily leverage medical codes while ClinicalBERT utilizes clinical notes to model the representations of EHR data.

**Baselines** are used to compare with NeuralCohort for fair evaluation. KNN, K-Means, DEC, DEKM, GRASP, DGLoS and IDC are adopted as baseline methods in our evaluation.

- **KNN** selects the nearest $K_N$ nodes for each node as its candidate neighbors where $K_N$ is the neighbor size in the Local Intra-cohort Modeling. After obtaining the initial representation $\mathcal{R}_0$, KNN is directly employed to identify the $K_N$ neighbors, which are then aggregated using the mean function to update the representations of the nodes.

- **K-Means** groups all the nodes into $|\mathcal{C}|$ clusters, where $|\mathcal{C}|$ represents the number of cohorts. The nodes within the same group share identical neighbors. Following the acquisition of the initial representation $\mathcal{R}_0$, K-Means is employed to form $|\mathcal{C}|$ clusters. Subsequently, a random selection of $K_N$ neighbors is performed for mean aggregation, leading to the update of the node's representation.

- **DEC** (Xie et al., 2016) is an unsupervised clustering method that jointly learns feature representations and cluster assignments using deep neural networks. DEC consists of two primary phases: pretraining and clustering. In the pretraining phase, DEC uses a deep autoencoder to initialize a low-dimensional latent space and during the clustering phase, DEC refines both the latent space and cluster assignments iteratively.

- **DEKM** (Guo et al., 2021) is a deep clustering method that trains an autoencoder to generate an embedding space, transforms the embedding space to a new space that reveals the cluster-structure information and optimizes the representation to increase the cluster-structure information in the new space.

- **GRASP** (Zhang et al., 2021) is designed to harness the information from similar patients through the representations of backbones via the Gumbel-Max technique and employs GCN with an inter-cohort graph to enhance representation learning. However, GRASP merely considers the relationship between cohorts, which limits its performance. In our evaluation, we implement GRASP using its open-source codes with the original settings adopted.

- **DGLoS** (Zang et al., 2023) implements a module grounded in graph representation learning to produce similarity-aware representations of patients, thereby

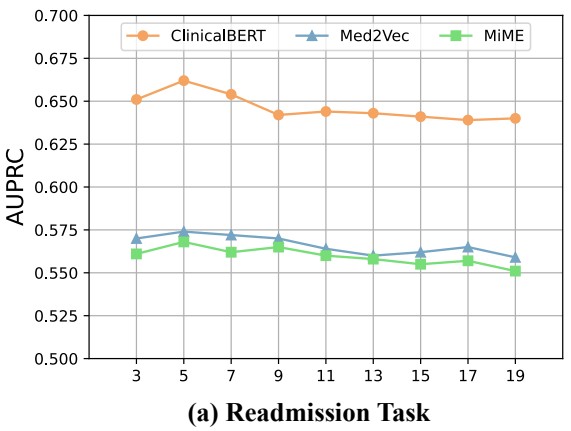
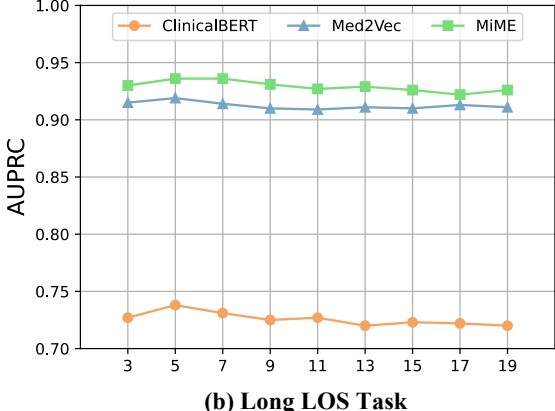

**(a) Readmission Task**    **(b) Long LOS Task**

*Figure 6.* Effects of hyperparameters $K_p$ for both tasks on the `MIMIC-III` dataset.

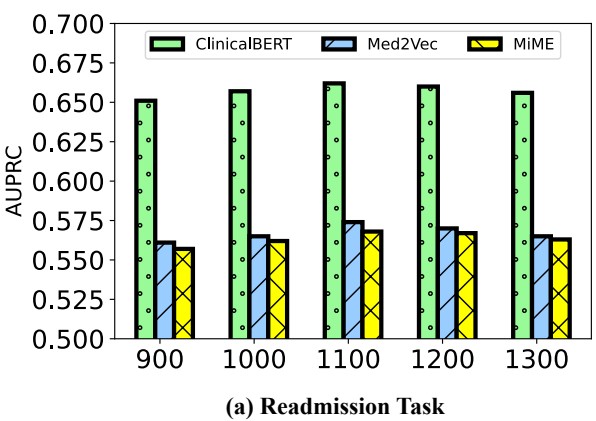
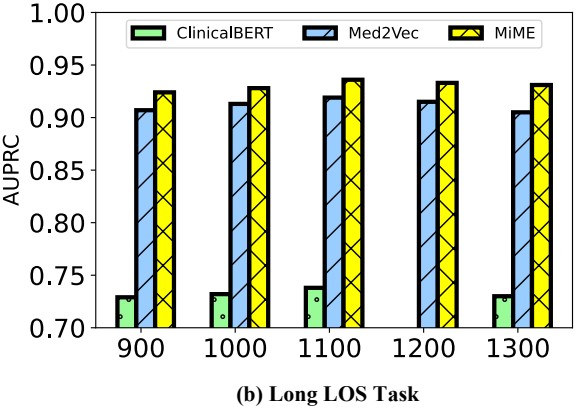

**(a) Readmission Task**    **(b) Long LOS Task**

*Figure 7.* Effects of the cohort numbers for both tasks on `MIMIC-III` dataset.

enhancing predictive accuracy. In DGLoS, patients are represented as nodes in the graph, with patient similarity as the weights of the edges. We adopt the deep modeling module, graph modeling module and prediction module with the backbone to achieve fair comparison.

- **IDC** (Svirsky & Lindenbaum, 2024) is a deep clustering model that accurately assigns clusters on tabular data. IDC employs a self-supervised approach to train an autoencoder and a gating network, enabling sample-specific feature selection by learning sparse, local feature representations that reconstruct data efficiently and then IDC refines cluster assignments using a clustering head optimized through a coding rate reduction loss to achieve compact and well-separated clusters.

## G. Experimental Settings

All the experiments are conducted on a server with Intel(R) Xeon(R) W-2133 CPU @ 3.60GHz, 64G memory, and 3 NVIDIA GeForce RTX 2080 Ti.

All datasets are pre-processed with similar logic to the experimental settings (Huang et al., 2019). We randomly split each dataset into training, validation, and test sets with a ratio of 8: 1: 1. For each model, we train 20 epochs (the models all converge within 20 epochs). The Pre-context Cohort Synthesis Module and Biscale Cohort Learning Module are trained in serial. In the Pre-context Cohort Synthesis Module, we employ icdcodex (Fisher, 2020) and GloVe (Pennington et al., 2014) as $g_p$ and $g_s$ in the Path and Semantics components of Hierarchical Visit Engine to distinguish similar paths in the ontology tree and capture the semantics, respectively. $f_D$, $f_M$, $f_L$, $f_{ML}$, $f_V$, $f_{attn}$ and $f_\phi$ are all standard MLP layers. For cohort derivation, K-Means is adopted to derive the initial centroid of each cohort. The dimensions of embedding for diagnosis, medication and

*Table 10.* Overall performance of NeuralCohort against baselines for readmission prediction on the `MIMIC-IV` dataset.

| Model | Readmission Task on `MIMIC-IV` | | |
|---|---|---|---|
| | **AUPRC** | **AUROC** | **Accuracy** |
| ClinicalBERT | 0.612±0.006 | 0.629±0.004 | 56.2%±0.4% |
| + KNN | 0.614±0.004 | 0.630±0.003 | 56.4%±0.2% |
| + K-Means | 0.614±0.004 | 0.629±0.003 | 56.4%±0.2% |
| + DEC | 0.619±0.006 | 0.635±0.002 | 56.5%±0.2% |
| + DEKM | 0.617±0.004 | 0.631±0.001 | 55.9%±0.2% |
| + GRASP | 0.626±0.005 | 0.641±0.004 | 56.9%±0.3% |
| + DGLoS | 0.623±0.002 | 0.619±0.004 | 57.1%±0.3% |
| + IDC | 0.627±0.007 | 0.643±0.003 | 57.5%±0.2% |
| **+ NeuralCohort** | **0.643±0.006** | **0.657±0.003** | **59.0%±0.4%** |
| Med2Vec | 0.561±0.007 | 0.617±0.004 | 55.4%±0.3% |
| + KNN | 0.560±0.004 | 0.618±0.004 | 55.3%±0.3% |
| + K-Means | 0.564±0.004 | 0.619±0.003 | 55.2%±0.2% |
| + DEC | 0.557±0.005 | 0.612±0.001 | 55.0%±0.3% |
| + DEKM | 0.562±0.007 | 0.618±0.005 | 55.7%±0.6% |
| + GRASP | 0.558±0.006 | 0.624±0.002 | 55.6%±0.5% |
| + DGLoS | 0.563±0.005 | 0.573±0.006 | 56.0%±0.5% |
| + IDC | 0.566±0.004 | 0.621±0.004 | 56.4%±0.5% |
| **+ NeuralCohort** | **0.579±0.004** | **0.639±0.005** | **57.6%±0.2%** |
| MiME | 0.572±0.005 | 0.624±0.004 | 55.6%±0.2% |
| + KNN | 0.571±0.003 | 0.619±0.004 | 55.4%±0.3% |
| + K-Means | 0.571±0.002 | 0.621±0.003 | 55.5%±0.2% |
| + DEC | 0.574±0.003 | 0.623±0.002 | 55.4%±0.4% |
| + DEKM | 0.570±0.002 | 0.619±0.004 | 55.2%±0.5% |
| + GRASP | 0.565±0.004 | 0.617±0.004 | 55.2%±0.3% |
| + DGLoS | 0.582±0.006 | 0.579±0.003 | 55.9%±0.2% |
| + IDC | 0.577±0.006 | 0.624±0.005 | 56.2%±0.3% |
| **+ NeuralCohort** | **0.604±0.005** | **0.641±0.006** | **57.9%±0.5%** |

*Table 11.* Overall performance of NeuralCohort against baselines for long LOS prediction on the `MIMIC-IV` dataset.

| Model | Long LOS Task on `MIMIC-IV` | | |
|---|---|---|---|
| | **AUPRC** | **AUROC** | **Accuracy** |
| ClinicalBERT | 0.650±0.004 | 0.578±0.003 | 58.1%±0.3% |
| + KNN | 0.654±0.003 | 0.581±0.005 | 58.1%±0.4% |
| + K-Means | 0.651±0.005 | 0.580±0.004 | 58.0%±0.4% |
| + DEC | 0.647±0.004 | 0.582±0.003 | 58.3%±0.2% |
| + DEKM | 0.662±0.003 | 0.589±0.004 | 59.2%±0.4% |
| + GRASP | 0.659±0.004 | 0.590±0.003 | 58.8%±0.5% |
| + DGLoS | 0.671±0.008 | 0.558±0.006 | 58.5%±0.4% |
| + IDC | 0.669±0.005 | 0.596±0.004 | 59.2%±0.2% |
| **+ NeuralCohort** | **0.724±0.003** | **0.635±0.002** | **61.8%±0.6%** |
| Med2Vec | 0.834±0.005 | 0.815±0.007 | 62.5%±0.6% |
| + KNN | 0.839±0.009 | 0.820±0.004 | 65.6%±0.3% |
| + K-Means | 0.831±0.004 | 0.814±0.004 | 61.5%±0.5% |
| + DEC | 0.827±0.003 | 0.810±0.002 | 62.7%±0.7% |
| + DEKM | 0.841±0.002 | 0.822±0.006 | 64.5%±0.4% |
| + GRASP | 0.839±0.004 | 0.824±0.004 | 63.0%±0.3% |
| + DGLoS | 0.837±0.003 | 0.784±0.003 | 64.2%±0.2% |
| + IDC | 0.842±0.006 | 0.823±0.008 | 65.8%±0.2% |
| **+ NeuralCohort** | **0.855±0.003** | **0.833±0.004** | **68.1%±0.3%** |
| MiME | 0.867±0.005 | 0.849±0.007 | 76.9%±0.4% |
| + KNN | 0.867±0.002 | 0.848±0.004 | 76.9%±0.2% |
| + K-Means | 0.869±0.004 | 0.850±0.003 | 76.8%±0.3% |
| + DEC | 0.867±0.002 | 0.856±0.004 | 77.2%±0.4% |
| + DEKM | 0.874±0.002 | 0.842±0.006 | 76.4%±0.4% |
| + GRASP | 0.873±0.003 | 0.859±0.004 | 77.5%±0.5% |
| + DGLoS | 0.872±0.004 | 0.837±0.002 | 78.2%±0.4% |
| + IDC | 0.873±0.002 | 0.853±0.004 | 77.3%±0.5% |
| **+ NeuralCohort** | **0.889±0.003** | **0.868±0.004** | **80.9%±0.2%** |

laboratory test, visit and patient is 128, 64, 64, 128 and 128. The dimensions of $\mathcal{R}_0$, $\mathcal{R}_L$, $\mathcal{R}_G$, and $\mathcal{R}_{final}$ are the same dimension as the setting of the backbone. $\lambda_{pcs}$, $\lambda_{JS}$ and $\lambda_{co}$ are set to 0.1 in our experiments. We use the Adam optimizer with an initial learning rate of $lr = 1e-3$. Hyperparameters $K_p = 5$, $|\mathcal{C}| = 1100$ remain consistent across all datasets and downstream tasks.

## H. Sensitivity Study

We conduct a sensitivity study to examine the impact of the key hyperparameters $|\mathcal{C}|$ and $K_p$ on the performance of NeuralCohort for both tasks using the `MIMIC-III` dataset, with all the backbones.

**Effects of Hyperparameter $K_p$.** In our investigation of the crucial hyperparameters' influence on NeuralCohort's performance, we first focus on the number of positive samples or negative samples $K_p$ in the Pre-context Cohort Synthesis Module. The AUPRC results, obtained by varying these hyperparameters using three datasets for both tasks, are illustrated in Figure 6. From this figure, we observe that the performance of NeuralCohort fluctuates at the beginning and subsequently decreases with the increase of $K_p$. When $K_p$

is excessively large, dissimilar patient pairs may be incorrectly identified as similar. Conversely, when $K_p$ is small, NeuralCohort fails to effectively learn representations due to insufficient maximization of Mutual Information Neural Estimation. Finally, these figures suggest that the optimal choice of $K_p$ is approximately 5, either increasing or decreasing these values will result in degraded performance.

**Effects of Cohort Number $|\mathcal{C}|$.** We evaluate the impact of cohort numbers on both tasks using the `MIMIC-III` dataset. The corresponding experimental results are depicted in Figure 7. It is clearly shown that the performance of all evaluated models first increases and then decreases as the number of cohorts increases. The reason for such phenomena is related to the target of NeuralCohort, *i.e.*, extracting cohort information for integration with backbones to achieve boosted performance. If the number of cohorts is smaller, it is harder for the Global Inter-cohort Modeling to capture the difference between the coarse-grained cohorts and if the size of cohorts tends to be larger, it is more likely to aggregate dissimilar patients, which leads to noisy information when utilizing the Local Intra-cohort Modeling. Conversely, with the number of cohorts increasing, fewer patients will be assigned to a cohort, which may

*Table 12.* Overall performance of NeuralCohort against baselines for readmission prediction on the `Diabetes130` dataset.

| Model | Readmission Task on Diabetes130 | | |
|---|---|---|---|
| | **AUPRC** | **AUROC** | **Accuracy** |
| Med2Vec | 0.623±0.004 | 0.628±0.003 | 59.4%±0.2% |
| + KNN | 0.625±0.003 | 0.625±0.004 | 59.2%±0.3% |
| + K-Means | 0.622±0.004 | 0.623±0.003 | 59.0%±0.2% |
| + DEC | 0.627±0.002 | 0.625±0.004 | 59.3%±0.4% |
| + DEKM | 0.624±0.004 | 0.631±0.003 | 59.7%±0.5% |
| + DGLoS | 0.629±0.005 | 0.614±0.007 | 59.9%±0.5% |
| + GRASP | 0.628±0.003 | 0.632±0.003 | 59.9%±0.4% |
| + IDC | 0.627±0.006 | 0.634±0.003 | 60.1%±0.3% |
| **+ NeuralCohort** | **0.641±0.003** | **0.644±0.004** | **61.5%±0.3%** |
| MiME | 0.657±0.002 | 0.662±0.004 | 62.7%±0.3% |
| + KNN | 0.649±0.003 | 0.657±0.004 | 62.1%±0.4% |
| + K-Means | 0.658±0.004 | 0.664±0.006 | 63.0%±0.2% |
| + DEC | 0.662±0.005 | 0.669±0.002 | 63.4%±0.4% |
| + DEKM | 0.658±0.002 | 0.665±0.003 | 62.7%±0.2% |
| + GRASP | 0.656±0.005 | 0.653±0.002 | 62.3%±0.5% |
| + DGLoS | 0.662±0.006 | 0.649±0.007 | 63.3%±0.2% |
| + IDC | 0.664±0.006 | 0.675±0.003 | 63.7%±0.4% |
| **+ NeuralCohort** | **0.676±0.006** | **0.689±0.004** | **65.4%±0.7%** |

*Table 13.* Overall performance of NeuralCohort against baselines for long LOS prediction on the `Diabetes130` dataset.

| Model | Long LOS Task on Diabetes130 | | |
|---|---|---|---|
| | **AUPRC** | **AUROC** | **Accuracy** |
| Med2Vec | 0.696±0.006 | 0.704±0.005 | 65.4%±0.5% |
| + KNN | 0.700±0.004 | 0.706±0.002 | 65.7%±0.4% |
| + K-Means | 0.701±0.003 | 0.709±0.004 | 65.8%±0.2% |
| + DEC | 0.700±0.003 | 0.702±0.004 | 65.7%±0.2% |
| + DEKM | 0.692±0.006 | 0.716±0.005 | 66.2%±0.4% |
| + GRASP | 0.693±0.003 | 0.700±0.003 | 65.1%±0.4% |
| + DGLoS | 0.705±0.003 | 0.681±0.006 | 66.0%±0.4% |
| + IDC | 0.717±0.007 | 0.714±0.004 | 66.5%±0.4% |
| **+ NeuralCohort** | **0.732±0.004** | **0.739±0.003** | **68.2%±0.2%** |
| MiME | 0.719±0.003 | 0.729±0.005 | 78.3%±0.4% |
| + KNN | 0.717±0.002 | 0.725±0.003 | 78.0%±0.2% |
| + K-Means | 0.720±0.004 | 0.732±0.005 | 78.6%±0.5% |
| + DEC | 0.724±0.008 | 0.739±0.004 | 79.4%±0.2% |
| + DEKM | 0.718±0.002 | 0.724±0.005 | 78.7%±0.2% |
| + DGLoS | 0.727±0.007 | 0.704±0.002 | 78.9%±0.4% |
| + GRASP | 0.724±0.005 | 0.731±0.005 | 78.7%±0.5% |
| + IDC | 0.732±0.005 | 0.740±0.002 | 79.6%±0.6% |
| **+ NeuralCohort** | **0.745±0.004** | **0.762±0.007** | **82.1%±0.3%** |

result in inadequate intra-cohort information. Therefore, the number of cohorts exerts a vitally important influence on both tasks using the `MIMIC-III` dataset, which means a suitable value should be selected for the number of cohorts. In our setting, the best $|\mathcal{C}|$ should be about 1100.

*Table 14.* Parameter Size and Running time of NeuralCohort.

| Model | Parameter Size (M) | Training Time (s) | Inference Time (ms) |
|---|---|---|---|
| ClinicalBERT | 109.4 | 4299.6 | 4.32 |
| + KNN | 109.4 | 6962.8 | 6.53 |
| + K-Means | 109.4 | 5525.2 | 6.15 |
| + DEC | 114.1 | 6225.2 | 5.95 |
| + DEKM | 115.4 | 6851.7 | 6.37 |
| + GRASP | 119.5 | 5292.4 | 7.29 |
| + DGLoS | 113.6 | 8418.8 | 7.93 |
| + IDC | 118.0 | 6546.2 | 7.04 |
| + NeuralCohort | 121.3 | 6432.4 | 7.16 |

## I. Experimental Results on `MIMIC-IV` and `Diabetes130` Datasets

Experimental results of `MIMIC-IV` and `Diabetes130` datasets are displayed in Table 10 to Table 13. It should be noted that there is no clinical text data in `Diabetes130` dataset, therefore, backbone ClinicalBERT cannot be employed in experiments with `Diabetes130` dataset. The results demonstrate that NeuralCohort consistently outperforms the baselines, confirming its effectiveness and generalizability.

## J. Efficiency Study

Table 14 compares the efficiency of NeuralCohort with backbone ClinicalBERT and baselines in terms of parameter size, training time, and inference time. NeuralCohort exhibits a comparable parameter size but achieves a reasonable training time, significantly outperforming computationally intensive methods such as DGLoS. In terms of inference time, NeuralCohort is competitive with DGLoS, GRASP and IDC but shows longer inference time than other methods. However, the trade-off in inference speed is balanced by the potential gains in modeling accuracy and representational power offered by NeuralCohort. Overall, the table demonstrates that NeuralCohort provides an effective balance between parameter complexity and computational efficiency, making it a promising choice for applications requiring cohort-aware learning.

## K. Extended Visualizaion

Extended visualization of all the methods is illustrated in Figure 8. The significant overlap among the points in Figure 8 (a) through (h) suggests that both the backbone and the baselines exhibit limited discriminative power. However, the incorporation of NeuralCohort substantially reduces this overlap, thereby enhancing the distinctiveness of the cohorts. This indicates that NeuralCohort effectively encodes the data into a representation similarity space.

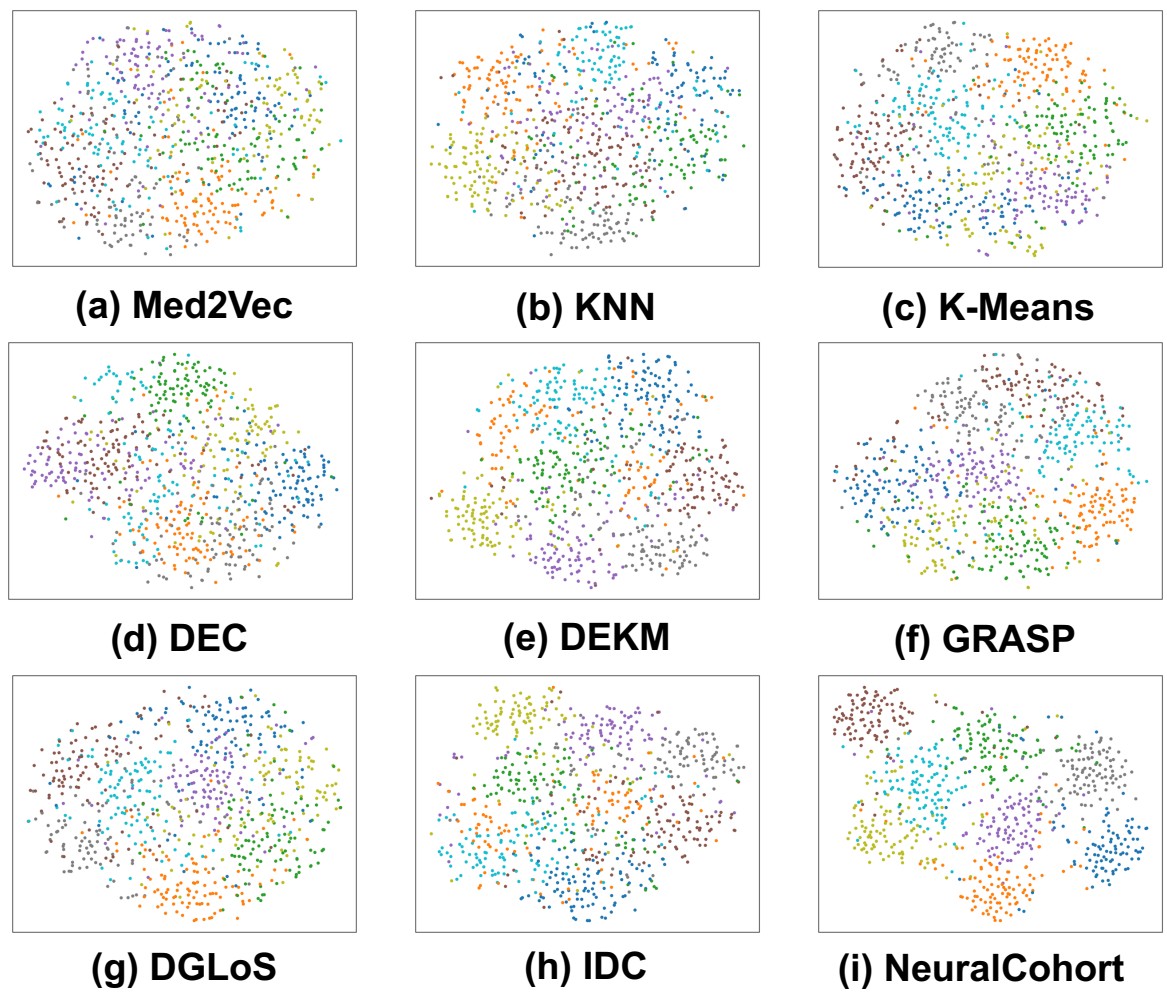

Figure 8. The extended t-SNE visualization of the final representations before prediction for selected eight-cohort points, where the backbone is Med2Vec.

