# OpenReview forum: "NeuralCohort: Cohort-aware Neural Representation Learning for Healthcare Analytics"
_ICML.cc/2025/Conference — ICML 2025 poster_

### Official Review · Reviewer_UU4H · 2025-03-08

**Overall Recommendation:** 2

**Summary:**

The paper proposes NeuralCohort, a cohort-aware neural representation learning method for healthcare analytics. It introduces two modules: (1) a Pre-context Cohort Synthesis Module to derive fine-grained cohorts via pseudo patient similarity, and (2) a Biscale Cohort Learning Module to integrate intra- and inter-cohort information. The method is evaluated on EHR datasets (MIMIC-III, MIMIC-IV, Diabetes130) for tasks like hospital readmission and length-of-stay prediction, demonstrating performance gains of up to 8.1% in AUROC when integrated with backbone models like ClinicalBERT and Med2Vec.

**Claims And Evidence:**

The claims are partially supported by experiments showing consistent improvements over baselines. However:

Problematic Claim: The assertion that NeuralCohort "simultaneously fulfills fine-grained cohort division and information exploitation" lacks direct validation. While ablation studies show performance drops when modules are removed, there is no qualitative analysis of cohort granularity or clinical interpretability.

Unjustified Choices: Hyperparameters (e.g., $\lambda_{p e s}$ =0.1) and cohort derivation methods (e.g., Jensen-Shannon divergence) are not rigorously justified. Sensitivity analyses for these parameters are missing.

**Essential References Not Discussed:**

See Relation To Broader Scientific Literature Above

**Experimental Designs Or Analyses:**

Verified: The ablation study (Table 3) confirms the contribution of each module. The consistent gains across datasets (Tables 1–2, 8–11) support generalizability.

Issues:

The cohort visualization (Figure 8) lacks clinical context (e.g., what distinguishes the cohorts in practice?).

**Methods And Evaluation Criteria:**

Methods: The hierarchical modeling of EHR data (diagnoses, medications, lab tests) and pseudo-similarity training are appropriate for temporal healthcare data. However, the cohort derivation process is under-detailed, and the reliance on pseudo-labels (without validation against expert annotations) is a limitation.

Evaluation: Standard tasks (readmission, LOS) and metrics (AUROC, AUPRC) are well-chosen. However, baselines like KNN and K-Means are outdated; comparisons with recent methods (e.g., contrastive learning or graph-based models) are missing.

**Other Comments Or Suggestions:**

N/A

**Other Strengths And Weaknesses:**

pros:

S1: Novel integration of cohort dynamics into neural representation learning.

S2: Modular design compatible with diverse backbones.

S3: Rigorous evaluation across multiple datasets and tasks.

cons:

W1: Limited novelty. This work only makes incremental modifications based on GRASP. I believe that GRASP's network architecture is also capable of utilizing both local intra-queue information and global inter-queue information, ultimately extracting comprehensive queue information.

W2: Lack of interpretability analysis for generated cohorts (e.g., clinical relevance).

W3: Incomplete baseline comparisons (e.g., state-of-the-art graph models).

W4: Insufficient details on cohort derivation and hyperparameter sensitivity.

**Questions For Authors:**

Q1: Were ROC-AUC improvements statistically significant (e.g., via bootstrapping or permutation tests)?

Q2: How were hyperparameters (e.g., learning rate, dropout) selected during fine-tuning?

Q3: How do confounding factors (e.g., medication use) explain inverse lab test correlations (e.g., LDL)?

**Relation To Broader Scientific Literature:**

The work connects to EHR representation learning (e.g., Med2Vec, ClinicalBERT) and patient similarity research. However, it overlooks recent advances:

Missing References: Temporal graph networks (e.g., TGN) for dynamic EHR modeling, contrastive learning frameworks (e.g., SimCLR) for patient similarity.

Gaps: No discussion of how NeuralCohort compares to dynamic cohorting methods (e.g., longitudinal clustering).

**Theoretical Claims:**

Lack of theoretical proof.

---

> ### Author Rebuttal · Authors · 2025-04-01
>
> Thanks for the detailed reviews. This is the link to our supplemented results: https://anonymous.4open.science/r/ICML2025-93F9.
>
> Q1: Lack validation of fine-grained cohorts. No qualitative analysis of cohort granularity or interpretability; visualizations lack clinical context.
>
> In practice, clinicians typically use coarse-grained cohorts (Sec. 4.4), making fine-grained grouping like that in Appx. K difficult to define manually. NeuralCohort enables the automatic discovery of such cohorts and identifies key features for clinical insight and decision support.
>
> Please refer to Appx. K, Q1, Q2 of Reviewer 6GpW and Q2 of Reviewer 8wHJ for the distinctive features of cohorts on three datasets.
>
> Q2: Hyperparameter and cohort derivation choices lack justification; no sensitivity analysis or tuning details.
>
> We conducted sensitivity analysis (Tables F and C), varying one hyperparameter at a time while keeping others fixed. JS divergence outperformed KL. The optimal values are: $\lambda_{pcs}$=0.1, $\lambda_{JS}$=0.1, $\lambda_{co}$=0.1, dropout=0.1, and learning rate=1e-3. When training, dropout is set to 0.1, and the setting of other parameters is shown in Appx. G.
>
> Q3: Cohort derivation lacks detail, and reliance on pseudo-labels without expert validation is a limitation.
>
> Please refer to Q3 of Reviewer 8wHJ for cohort derivation details. Pseudo-labeling is widely used in representation learning[1], especially in EHR, where manual labeling is costly and impractical. Expert-defined similarity requires extensive clinical review, which is challenging at scale. Our method remains flexible and can incorporate expert input in future semi-supervised or clinician-in-the-loop settings.
>
> Q4: Baselines are outdated; recent methods and references are missing. No comparison to temporal graph models, contrastive learning, or dynamic cohorting approaches.
>
> We briefly discuss related methods and will expand in revision. TRANS[2] models  EHR as a temporal heterogeneous graph to capture both temporal and structural information. SimSig[3] uses a contrastive learning approach to learn similar embeddings of patients with physiological signal data. Longitudinal K-Means[4] presents K-Means approaches for subtyping opioid use trajectories from EHR data and interprets the resulting subtypes using decision trees.
>
> We clarify that temporal graph networks for EHR modeling are complementary to our method and could serve as backbones rather than baselines. Comparisons with SimSig and Longitudinal K-Means in Table G show they underperform our method.
>
> Q5: Limited novelty; the method appears to be an incremental extension of GRASP, which may already capture both local and global cohort-level information.
>
> Unlike GRASP's flat KNN graph, NeuralCohort introduces a distinct, bi-level architecture that explicitly models both intra-cohort and inter-cohort structures through dedicated modules. This enables fine-grained cohort discovery and more effective cohort-aware representation learning. Unlike GRASP, NeuralCohort integrates medical semantics into the learning process, which our experiments show leads to more stable performance—whereas GRASP may degrade backbone performance due to less clinically grounded similarity signals.
>
> Q6: Were ROC-AUC improvements statistically significant?
>
> We evaluate AUROC difference with MIMIC-III dataset on long LOS task in Table H, showing statistically significant improvements.
>
> Q7: How do confounding factors explain inverse lab test correlations?
>
> Counterintuitive correlations, such as low LDL linked to worse outcomes, may result from confounding factors like medication use. Patients with low LDL often have prior cardiovascular risk and are treated with statins, making low LDL a marker of treated high-risk status rather than low risk. NeuralCohort captures such latent clinical structures by modeling intra-cohort similarities and inter-cohort differences. For instance, it can distinguish between statin-treated high-risk patients and untreated low-risk individuals, offering more clinically meaningful stratification than models treating patients independently. Please refer to Appx. K, Q1, Q2 of Reviewer 6GpW and Q2 of Reviewer 8wHJ for practical applications.
>
> [1] Xi, Liang, et al. "Semi-supervised time series classification model with self-supervised learning." Engineering Applications of Artificial Intelligence 116 (2022): 105331.
>
> [2] Chen, Jiayuan, et al. "Predictive modeling with temporal graphical representation on electronic health records." IJCAI: proceedings of the conference. Vol. 2024. 2024.
>
> [3] Shanto, Subangkar Karmaker, et al. "Contrastive Self-Supervised Learning Based Approach for Patient Similarity: A Case Study on Atrial Fibrillation Detection from PPG Signal." arXiv preprint arXiv:2308.02433 (2023).
>
> [4] Mullin, Sarah, et al. "Longitudinal K-means approaches to clustering and analyzing EHR opioid use trajectories for clinical subtypes." Journal of biomedical informatics 122 (2021): 103889.

---

### Official Review · Reviewer_8wHJ · 2025-03-12

**Overall Recommendation:** 3

**Summary:**

This paper introduces the NeuralCohort framework, a novel method for neural cohort generation and selection based on two distinct strategies that model and generate (local) intra- and (global) inter-cohort information using a pre-context synthesis module employing pseudo similarity and a biscale cohort learning module to fuse and predict downstream classification tasks.

**Claims And Evidence:**

The authors provide interesting experiments and ablation studies for NeuralCohort. Firstly, they claim and demonstrate that NeuralCohort can consistently perform with different backbone neural architectures in their experiments and can outperform current state-of-the-art embedding/clustering methods for two classification tasks, namely "cross-visit Hospital Readmission Prediction" and "within-visit Long
Length-of-Stay". This empirically supports their claim that NeuralCohort is a strong encoding/feature aggregation method.

**Essential References Not Discussed:**

I have not found any relevant papers that the authors have missed.

**Experimental Designs Or Analyses:**

The authors provide two experiments on three different datasets with three different backbones, which makes the case that their method works reliably and is adaptable to other backbones.
As mentioned previously, it is hard to understand from the main paper/appendix which dimensions feature spaces span, how the authors derive the number of clusters for their experiments, and how architectural design choices (e.g. cluster number) affect the baseline methods. I have read the interesting ablation study in the Appendix. Still, to me, it is not ultimately clear how the baseline methods were tuned (if they were tuned) and how robust the baselines are across a parameter set.

While the paper claims to provide a method for cohort generation, it is validated on two downstream segmentation tasks that serve as a proxy for this problem. While I am unaware of a better experiment/task to demonstrate it, I believe this is a bit suboptimal to support the claims.

**Methods And Evaluation Criteria:**

All architectural design considerations are properly introduced with mathematical formulation. While the formulations fundamentally make sense, some of the decisions "Hierarchical Visit Engine" or "PseudoSim Training" are not adequately motivated. It is hard to understand what other possibilities have been tested in related literature and why the authors believe that this constitutes the best and/or most robust method to model the discussed concept.  For instance, how would a contrastive learning approach such as SimCLR [1] help or impede the modeling compared to the PseudoSim Training? For instance, it is unclear to me if this architectural design choice is something that works well for the two datasets used in this study or if it is clinically meaningful in other studies, such as Multiple Sclerosis research. I also find it very hard to understand - based on the manuscript and not the codebase - the intermediate steps of the framework and the dimensionality of different feature spaces. This makes it hard to assess if the evaluation of experiments is fair and how baseline algorithms have been tuned or if the experimental design is posed in a way that favors NeuralCohort.

[1] Chen T, Kornblith S, Norouzi M, Hinton G. A simple framework for contrastive learning of visual representations. InInternational conference on machine learning 2020 Nov 21 (pp. 1597-1607). PmLR.

**Other Comments Or Suggestions:**

N/A

**Other Strengths And Weaknesses:**

Strengths:
- I would like to highlight that the authors have already shared their code in the reviewing phase.

Weaknesses:
- To me this paper lacks details regarding how to use NeuralCohorts for other datasets and the intended use case - the formation and selection of cohorts in studies.

**Questions For Authors:**

- How do you typically select hyperparameters such as a positive number of pairs for PseudoSIM for Neural Cohorts?
- How were they selected for this paper's experiments? Given the variability of the ablation study in the appendix, are the results in the reported tables the best-case scenario?
- Were the number of clusters empirically selected for Figure 8, Supplementary Material? How do the plots look for a different number of clusters ? Given that the performance of NeuralCohort in your experiments is better than baselines but not always by a high margin, I am surprised to see such distinct cluster in comparison.
- Did you assess features in the neural cohort clusters? Are they medically meaningful, which would be significant considering that it should be used in medical cohorts?

**Relation To Broader Scientific Literature:**

The authors claim that NeuralCohort is (one of the) first methods to look into cohort selection apart from traditional methods used by clinicians and in clinical studies. They correctly set their work into the context of neural clustering algorithms, which they also use as baselines. I feel the methods section could explain more about why the authors have decided to model certain relationships the way they have decided to do and how they ground this in prior work.

**Theoretical Claims:**

The manuscript includes no novel proofs but mathematical formulations of architectural design choices.

---

> ### Author Rebuttal · Authors · 2025-04-01
>
> We appreciate your insightful comments. This is the link to our supplemented results: https://anonymous.4open.science/r/ICML2025-93F9.
>
> Q1: The possibilities of techniques. How would SimCLR help?
>
> Key components like reverse-time attention and mutual information have been validated in prior work (e.g., RETAIN[1], FASTDeepHit[2]). While other architectures are possible, our choice achieves SOTA performance, generalization, and interpretability. Many of the components are orthogonal to our primary contribution so NeuralCohort can readily incorporate more advanced alternatives as they become available.
>
> SimCLR, where we randomly mask EHR to construct positive pairs, degrades performance as shown in Tables D and E. The reason lies in SimCLR is designed for image data, where augmentations like cropping retain semantic meaning. In EHR, creating label-preserving augmentations is more challenging considering the complex medical semantics of EHR.
>
> Q2: Method only works well for the two datasets? How to use NeuralCohorts for other datasets?
>
> NeuralCohort is tailored for EHR data and suitable for domains where subpopulation patterns are important, such as diabetes or Multiple Sclerosis. We evaluate NeuralCohort on Diabetes130 dataset (Appx. I), with clinical insights detailed in Q2 of Reviewer 6GpW.
>
> We have done the clinical application on Multiple Sclerosis dataset. The distinctive features are listed in Table B.
>
> Cohort #1 shows early neurological abnormalities. They should be prioritized for early initiation of DMTs and placed on close clinical surveillance schedules. Hospitals and neurology departments should ensure these patients have expedited access to high-resolution neuroimaging, neurologist consultations, and structured follow-up. Cohort #2 shows slower progression and may follow atypical paths. A conservative management approach with lifestyle interventions is suitable. Operationally, they can be monitored in outpatient settings with less frequent MRI, reserving resources for higher-risk patients. Patients in Cohort #3 display minimal abnormalities, indicating a stable, low-risk, or non-converting group. The optimal strategy for managing this cohort involves conservative care focused on patient reassurance, symptom education, and minimal intervention unless clinical deterioration is observed. Cohort #4 includes younger patients with positive LLSSEP and VEP, suggesting early subclinical neurological involvement despite low EDSS scores. This profile signals elevated risk for MS conversion. Clinical strategies should focus on prevention through lifestyle counseling, low-risk DMTs, and regular monitoring. Besides, this group warrants access to early neurology resources and is well-suited for clinical trials aimed at delaying progression before disability develops.
>
> Q3: Which dimensions feature span, how to derive the number of clusters, the number of pairs, and how the baselines were tuned?
>
> The dimensions of embedding for diagnosis, medication and laboratory test, visit and patient are 128, 64, 64, 128 and 128. The dimensions of $\mathcal{R} _ 0$, $\mathcal{R} _ L$, $\mathcal{R} _ G$, $\mathcal{R} _ {final}$ are the same dimension as the setting of the backbone.
>
> We use Student's t-distribution as a soft assignment function to map patient to cohort, computing similarity via a t-kernel to produce normalized soft assignments. The number of clusters determines the number of centroids and is initialized using KMeans. A sensitivity study on cohort number and pair numbers is provided in Appx. H. All experiments are repeated five times, and the average results are reported.
>
> The following protocol is used to ensure fair comparisons. We followed the original papers' recommended hyperparameters or adopted defaults from the official implementation when tuning details were unavailable. To ensure fairness, shared hyperparameters—such as the number of cohorts or clusters—were kept consistent across all methods.
>
> Q4: It is a bit suboptimal to support the claims.
>
> We explicitly address four key aspects: improved task performance (Sec. 4.2, Appx. I), CH-score for similarity evaluation (Sec. 4.5), t-SNE visualizations of cohort (Sec. 4.5), and clinical validation on MIMIC-III (Appx. K) and Diabetes130 (Q2, Reviewer 6GpW), which could validate the generated cohorts.
>
> Q5: How does Figure 8 look for a different number of clusters?
>
> The number of clusters is chosen to support clear visualization. Below, we present results for 6 and 10 clusters in Figure A and Figure B. While some baselines form clusters, they lack clear boundaries, whereas NeuralCohort produces more distinct groupings.
>
> [1] Choi, Edward, et al. "Retain: An interpretable predictive model for healthcare using reverse time attention mechanism." Advances in neural information processing systems 29 (2016).
>
> [2] Do, Hyungrok, et al. "Fair survival time prediction via mutual information minimization." Machine Learning for Healthcare Conference. PMLR, 2023.

---

### Official Review · Reviewer_6GpW · 2025-03-13

**Overall Recommendation:** 2

**Summary:**

This paper proposes NeuralCohort, a cohort-aware neural representation learning method for healthcare analytics. The approach segments patients into fine-grained cohorts and captures both intra- and inter-cohort information through a two-module paradigm.

**Claims And Evidence:**

The paper presents a well-supported evaluation of its method using multiple EHR datasets, demonstrating performance improvements. However, it lacks clear evidence on how cohort insights translate into clinical interventions.

**Essential References Not Discussed:**

NA

**Experimental Designs Or Analyses:**

The experimental setup has no significant methodological flaws identified. However, the study would benefit from demonstrating practical applications beyond the MIMIC dataset.

**Methods And Evaluation Criteria:**

The methodology integrates existing techniques (hierarchical modeling, GNNs, contrastive learning) without introducing fundamentally new innovations.

**Other Comments Or Suggestions:**

NA

**Other Strengths And Weaknesses:**

The paper's strength lies in its comprehensive evaluation across multiple EHR datasets, demonstrating consistent performance improvements when integrated with existing backbone models.

I do not find significant methodological or experimental flaws in the study. However, my primary concern is the novelty of the approach. The method appears to be an integration of existing techniques—hierarchical modeling, graph neural networks, and contrastive learning—rather than introducing fundamentally new methodological innovations. Both the core idea and the task itself are not particularly novel compared to existing work, especially given the extensive exploration of similar tasks on the MIMIC dataset.

Additionally, it remains unclear how the proposed work and the insights derived from cohort segmentation translate into tangible clinical interventions or improved patient management strategies. I suggest that the authors clearly articulate what is conceptually novel beyond the integration of existing techniques. Moreover, providing concrete examples of how the identified cohorts could inform specific clinical decisions or extending the application beyond standard prediction tasks—already well-studied on MIMIC datasets—would strengthen the contribution of this work.

**Questions For Authors:**

NA

**Relation To Broader Scientific Literature:**

NA

**Theoretical Claims:**

NA

---

> ### Author Rebuttal · Authors · 2025-04-01
>
> Thanks for your constructive feedback. This is the link to our supplemented results: https://anonymous.4open.science/r/ICML2025-93F9.
>
> Q1: Lack clear evidence on how cohort insights translate into clinical interventions.
>
> Please refer to Appx. K for detailed analysis. To be more specific, Cohort #1 includes patients with cardiovascular conditions and is at high risk for acute cardiac events and extended hospital stays[1]. Early identification enables the prioritization of telemetry beds, cardiology consults, and monitoring. Clinically, this enables timely diuretics, echocardiography, and discharge planning with heart failure follow-up. Cohort #2 features patients with chronic metabolic and hematologic conditions that often co-occur and need interdisciplinary care. Identifying this cohort allows hospitals to mobilize diabetes educators, lipid clinics, and anticoagulation monitoring services. Clinically, it enables insulin titration, lipid therapy, and bleeding risk management to reduce complications and improve outcomes. Cohort #3 is characterized by renal and urological issues requiring frequent labs, fluid monitoring, and nephrology support. Early identification allows hospitals to allocate renal panels, schedule imaging for uropathy, and prepare dialysis resources—helping manage AKI risk and avoid care escalation. Cohort #4 presents complex chronic and acute conditions, requiring coordination across pulmonology, nephrology, endocrinology, and infectious disease. Identifying this cohort supports planning for respiratory support, hormone therapy, infection control, and opioid withdrawal. Operationally, it enables resource allocation for respiratory teams, isolation rooms, and endocrine/renal labs.
>
> Q2: Practical applications beyond the MIMIC dataset
>
> We evaluated NeuralCohort on a real-world Diabetes130 dataset, which is shown in Appx. I. We also analyze the practical application, similar to Appx. K but on the readmission task. The distinctive features are shown in Table A.
>
> Cohort #1 is characterized by longer hospital stays, insulin use, and frequent prior outpatient visits, indicating complex or unstable diabetes requiring intensive inpatient care. These patients often need time to stabilize their glucose levels, adjust medications, and manage comorbid conditions. They benefit from diabetes management plans, glucose monitoring, and endocrinology support. Hospitals should allocate extended-stay beds, involve diabetes educators, and initiate early discharge coordination. Close follow-up via outpatient care or telemedicine helps reduce readmission risk. Cohort #2 comprises low-acuity patients. Likely admitted for scheduled care or monitoring, they benefit from preventive care, education, and routine screening. Hospitals can streamline care using standardized protocols, fast-track discharge workflows, and minimal specialty consults, optimizing resource use for this stable cohort. Defined by a high rate of emergency admissions, younger age, and few prior inpatient visits, cohort #3 likely represents underserved or poorly engaged patients who rely on the ER for primary care needs. They may face social or behavioral health challenges. Interventions should focus on screening for social determinants, community referrals, and behavioral health support. Hospitals can prioritize care navigation and ER diversion efforts to reduce unnecessary admissions and alleviate emergency departments burden. Cohort #4 consists of high-risk, high-utilization patients. Tailored interventions should focus on intensive transitional care management, including medication reconciliation, post-discharge follow-ups within 48–72 hours, and multidisciplinary care planning. Hospitals can assign care managers, enroll patients in chronic care programs, and coordinate outpatient services to reduce readmissions and enhance long-term outcomes.
>
> Q3: Novelty of the approach.
>
> It should be noted that NeuralCohort is an application-driven ML submission targeting practical healthcare applications, where originality could mean "a novel combination of existing methods to solve the task at hand so as to match the needs of the user" rather than wholly novel methods[2]. Our key innovation lies in the cohort-aware representation learning framework, which explicitly disentangles and models both local intra-cohort and global inter-cohort structures — a perspective not addressed in prior work. Moreover, NeuralCohort not only enhances predictive performance but also yields clinically interpretable embeddings that align with real-world needs, such as intervention planning and resource optimization.
>
> [1] Tigabe Tekle, Masho, Abaynesh Fentahun Bekalu, and Yonas Getaye Tefera. "Length of hospital stay and associated factors among heart failure patients admitted to the University Hospital in Northwest Ethiopia." Plos one 17.7 (2022): e0270809.
>
> [2] https://icml.cc/Conferences/2025/ReviewerInstructions

---

> > ### Comment · Reviewer_6GpW · 2025-04-05
> >
> > Thanks to the authors for their reply. I would stand for my score due to the limited novelty of the clinical tasks and insights.

---

> > > ### Author Response · Authors · 2025-04-09
> > >
> > > We sincerely thank the reviewer for further feedback. Below, we provide evidence and results to further clarify the insights derived from the cohorts. The observed **differences in average length of stay (LOS)** across cohorts, along with **statistically significant p-values**, support the presence of **meaningful distinctions among them**. **A clinical specialist** was consulted to provide guidance from a clinical research perspective, and has **validated the distinctive cohort features, the associated intervention strategies, and the implications for resource optimization** in Q1 on MIMIC-III dataset and Q2 on Diabetes130 dataset and Q2 of reviewer 8wHJ on Multiple Sclerosis dataset.
> > >
> > >
> > > | Cohort Index | Avg. LOS (days) | Distinctive Features            |p-value|
> > > | ------------ | --------------- | ------------------------------- |-|
> > > | Cohort #1    | 11.2            | HF, Arrhythmia, CHD, Angina |<0.001|
> > > | Cohort #2    | 6.4             | Diabetes, DLM, CD, Obesity      |<0.001|
> > > | Cohort #3    | 8.9             | OUA, PSL, UF, F&amp;E Disorders |0.0012|
> > > | Cohort #4    | 5.1             | OPI, CKD, Thyroiditis, Flu      |<0.001|
> > >
> > >
> > >
> > > Thanks for the feedback on the novelty. We would like to provide the following clarification to better highlight the novel contributions of our work.
> > >
> > > - We propose a general cohort-aware neural representation learning method, NeuralCohort to address **the overlook of fine-grained cohorts and comprehensive cohort information exploitation.**
> > > - To the best of our knowledge, **we are the first to explicitly model both intra-cohort and inter-cohort information based on pseudo-medical-similarity semantics** and encode them into augmented representations for downstream prediction tasks.
> > > - NeuralCohort is designed to be **a flexible and modular plug-in** that can be seamlessly integrated with various backbone models, enhancing their ability to capture cohort-specific insights and improve predictive performance.
> > > - We evaluate NeuralCohort on three real-world EHR datasets across two tasks and analyze the clinical insights for the generated cohorts.  The experimental results demonstrate its **effectiveness**, and the cohort insights—validated by a clinical specialist—highlight its **clinical interpretability and significance**.

---

### Official Review · Reviewer_uWw6 · 2025-03-14

**Overall Recommendation:** 4

**Summary:**

The paper proposes NeuralCohort -- a cohort-aware neural representation learning method designed to improve electronic health record (EHR) analysis. It addresses the challenges of fine-grained cohort segmentation and effectively utilizes both intra- and inter-cohort information. By incorporating the Pre-context Cohort Synthesis for pseudo patient similarity prediction task, and Biscale Cohort Learning modules to learn the intra- and inter- cohort interactions, NeuralCohort enhances existing models, boosting their performance. Its approach not only improves predictive accuracy but also provides valuable medical insights for EHR studies.

**Claims And Evidence:**

Yes

**Essential References Not Discussed:**

n/a

**Experimental Designs Or Analyses:**

The experiments are well designed to showcase the effectiveness of the approach. The suuplementary material has more addendum experiments.

**Methods And Evaluation Criteria:**

Yes, the methods had been extensively tested on popular MIMIC-III, MIMIC-IV and Diabates130 datasets which are benchmarks for the EHR domain.

**Other Comments Or Suggestions:**

n/a

**Other Strengths And Weaknesses:**

Strengths
- The ideas of modeling in 2 stage manner - first visit level then modeling the intra- and inter- cohort signals are novel.
- The method is clearly effective and also interpretible which is a key consideration for healthcare domain.

**Questions For Authors:**

n/a

**Relation To Broader Scientific Literature:**

Prior works have not sufficiently explored the cohort formulation for EHR tasks. Patient similarity has been tackled based on expert labellings and more recently cosine similarity between embeddings from EHR data. This work models patient's visit level representation to then model cohort interactions. It shows the worth of cohort studies. This could be useful for most downstream healthcare analysis tasks.

**Theoretical Claims:**

Yes mostly, did not go into details.

---

> ### Author Rebuttal · Authors · 2025-04-01
>
> Thank you for your thoughtful and encouraging review. We sincerely appreciate your recognition of NeuralCohort as a cohort-aware neural representation learning framework that effectively addresses the challenges of fine-grained cohort segmentation and the modeling of both intra- and inter-cohort relationships. We are especially pleased that you found the two-stage design — modeling at the patient visit level followed by biscale cohort learning — to be both novel and interpretable. Our goal is to not only improve predictive performance across benchmark EHR datasets such as MIMIC-III, MIMIC-IV, and Diabetes130, but also to provide clinically meaningful insights through interpretable cohort structures. Your support affirms the value of our approach for advancing cohort-based analysis in real-world healthcare applications.

---

> > ### Comment · Reviewer_uWw6 · 2025-04-06
> >
> > n/a

---

### Decision · Program_Chairs · 2025-05-01

**Decision:**

Accept (poster)

**Comment:**

This paper introduces NeuralCohort, a cohort-aware neural representation learning for electronic health record (EHR) analysis. The method integrates cohort segmentation through two modules: the Pre-context Cohort Synthesis and the Biscale Cohort Learning Module. Evaluations on diverse EHR datasets (MIMIC-III, MIMIC-IV, Diabetes130) demonstrate notable improvements (up to 8.1% AUROC gains) when integrated with baseline models such as ClinicalBERT and Med2Vec.

Reviewers agree that this paper provide a novel integration of cohort dynamics into neural representation learning and has solid evaluation across multiple EHR datasets and prediction tasks.